# Perceptual metacognition of human faces is causally supported by function of the lateral prefrontal cortex

Regina C. Lapate [1,2,3 ✉], Jason Samaha [4], Bas Rokers [5,6], Bradley R. Postle[5,7] & Richard J. Davidson [3,5,7]

Metacognitive awareness—the ability to know that one is having a particular experience—is thought to guide optimal behavior, but its neural bases continue to be the subject of vigorous debate. Prior work has identified correlations between perceptual metacognitive ability and the structure and function of lateral prefrontal cortex (LPFC); however, evidence for a causal role of this region in promoting metacognition is controversial. Moreover, whether LPFC function promotes metacognitive awareness of perceptual and emotional features of complex, yet ubiquitous face stimuli is unknown. Here, using model-based analyses following a causal intervention to LPFC in humans, we demonstrate that LPFC function promotes metacognitive awareness of the orientation of faces—although not of their emotional expressions. Collectively, these data support the causal involvement of the prefrontal cortex in metacognitive awareness, and indicate that the role of LPFC in metacognition encompasses perceptual experiences of naturalistic social stimuli.

[1] Helen Wills Neuroscience Institute, University of California, Berkeley, Berkeley, CA, USA. [2] Department of Psychological and Brain Sciences, University of California, Santa Barbara, Santa Barbara, CA, USA. [3] Center for Healthy Minds, University of Wisconsin-Madison, Madison, WI, USA. [4] Department of Psychology, University of California, Santa Cruz, Santa Cruz, CA, USA. [5] Department of Psychology, University of Wisconsin-Madison, Madison, WI, USA. [6] Department of Psychology, New York University Abu Dhabi, Abu Dhabi, United Arab Emirates. [7] Department of Psychiatry, University of Wisconsin-Madison, Madison, WI, USA. ✉email: lapate@ucsb.edu

When navigating their complex social environments, humans can often monitor and report on their thoughts, feelings, and experiences—albeit not without error[1]. The ability to introspect on one's own mental content is termed metacognition, a process that gives rise to representations considered important for conscious experiences[2]. Metacognition is thought to guide decision making—for instance, lower levels of metacognitive awareness have been observed in individuals espousing radical beliefs[3] and in various forms of psychopathology[4,5]. Yet there continues to be vigorous debate regarding the neural architecture that gives rise to metacognitive ability.

Prominent theories of consciousness, such as the Global Workspace Theory and Higher Order Theories[6,7], posit a relationship between metacognition and conscious perception. For instance, metacognitive awareness, the ability to accurately monitor one's internal experiences, has been proposed to be a precursor to consciousness[8]. Relatedly, some argue that all conscious percepts may be inherently imbued with a metacognitive component, which would facilitate their integration in a common global workspace and permit optimal decision making across domains[6] (but see also ref. [7]). While some of these theories disagree on the precise relationship between metacognition and consciousness, they converge in proposing that function of the prefrontal cortex (PFC) plays a critical role in promoting metacognitive awareness and conscious perception, putatively regardless of whether the content of conscious awareness is explicitly reported[9,10] (but see also ref. [11]).

Research on the neural substrates of visual metacognition largely agrees with the above-mentioned theories, and indicates that function and structure of anterior and dorsal regions of the lateral prefrontal cortex (LPFC) often correlate with metacognitive awareness[12–18]. However, there remains considerable controversy as to whether studies using methods that permit causal inference—such as lesion and brain stimulation studies—actually support a causal role for LPFC in metacognition[19–23]. Moreover, extant work has primarily examined metacognition of low-level visual features using largely nonsocial stimuli (e.g., Gabor patches, dots, or geometric shapes; for an exception see ref. [24])—even though accurate metacognitive awareness should be particularly crucial for adaptive behavior when individuals encounter complex (and often ambiguous) sources of motivationally relevant information, such as human faces.

Metacognitive awareness is quantified by examining the trial-by-trial correspondence between objective performance (i.e., stimulus-discrimination accuracy) and the observer's subjective reports on the clarity of—or confidence in—their perceptual experience[25]. As the correspondence between objective and subjective reports increases (e.g., when high-confidence ratings follow correct trials, and low-confidence ratings follow incorrect trials), metacognitive awareness approaches maximum. Across individuals, greater metacognitive ability in the visual domain correlates with gray matter volume in anterior PFC[14,17], gray matter myelination in dorsal LPFC (DLPFC)[26], and white matter microstructure of prefrontal fibers[17,27]. Accordingly, patients with lesions to LPFC show visual metacognition deficits despite having intact stimulus-discrimination performance[28], and require longer stimulus presentations to subjectively report visual experiences[29] (but see also ref. [20]).

In a seminal demonstration of the putative causal role of LPFC in promoting metacognition in healthy adults, Rounis and colleagues (2010) altered LPFC function during a simple-shape two-choice discrimination task using an inhibitory transcranial magnetic stimulation (TMS) protocol, continuous theta-burst (cTBS)[16]. Their results indicated that inhibitory cTBS to LPFC impaired metacognitive awareness of simple geometric shapes while sparing objective discrimination performance. However, subsequent TMS studies on the causal status of LPFC for visual metacognition have provided mixed results: Bor et al. (2017) failed to replicate Rounis et al. (2010)[21]—but, neither study used neuroanatomically-guided navigation during the administration of TMS—thus, whether those studies targeted the same LPFC region across participants is unclear. Other experiments using anatomically and/or functionally guided TMS interventions have offered support for causal contributions of DLPFC and anterior PFC in shaping confidence and metacognition, respectively[30,31]—however, those findings have not always been consistent in directionality with previously-obtained results[16]. Thus, additional work is required to clarify the nature and causal status of LPFC function in visual metacognition.

Determining the real-world import of LPFC function in visual metacognition—as well as its potential limits—necessitates the adoption of ecologically relevant, complex stimuli that are ubiquitous in everyday life, such as human faces. This approach, coupled with precise TMS neuronavigation, may not only help adjudicate between prior disparate findings, but also help clarify domain specificity in the neural architecture supporting metacognition. For instance, while metacognition of low-level visual features often correlates with function of LPFC[12] (see ref. [32] for a meta-analysis), it remains unclear whether metacognition of emotional features relies on the same lateral prefrontal network[33] or whether it may instead rely on a separate medial prefrontal, interoceptive-representing circuitry[24,34].

Therefore, in this study, we tested whether LPFC function plays a causal role in metacognitive awareness of human face stimuli. To temporarily manipulate LPFC function, we administered an inhibitory TMS protocol (cTBS) to LPFC as well as to a control site (somatosensory cortex; S1) in a within-subjects design. The administration of TMS was conducted using neuronavigation as to accurately and consistently localize LPFC and control targets based on neuroanatomical landmarks (T1-weighted scan) for each subject (Fig. 1a).

Following a 20-s cTBS protocol (Fig. 1b), observers performed two face-discrimination tasks. Because prior relevant work on visual metacognition had often probed metacognition of perceptual decisions involving stimulus orientation[16,21,30,31], we asked observers to discriminate the orientation of emotional faces (Face Orientation task; upright vs. upside down) (Fig. 1c). A separate, secondary task assessed potential domain specificity in the role of LPFC in visual metacognition by going beyond low-level features, and examining participants' metacognition of the emotional content of the faces—to do so, we asked observers to discriminate between emotional expressions (Face Emotion task; happy vs. fearful) (Fig. 1d). In both tasks, observers performed two-choice stimulus discriminations of fearful and happy faces, which were followed by subjective reports on the clarity of (confidence in) their visual experience (using the 4-point scale Perceptual Awareness Scale[35]). Emotional faces were presented at six different contrasts using the method of constant stimuli (Fig. 1c inset), which allowed us to examine whether the putative role for LPFC in promoting metacognitive visual awareness was specific to near-threshold stimuli presentations (as reported previously, e.g., ref. [16]), or whether it extended to sub- and supra-threshold stimuli.

We assessed metacognitive awareness using three distinct metrics frequently employed in the literature (for details, see Methods): First, using the robust and nonparametric Type 2 Area Under the Receiver Operating Characteristic Curve (Type 2 AUC), which indexes metacognitive sensitivity. Because metacognitive sensitivity can be influenced by fluctuations in task performance, we also used a Bayesian model[36] to compute the metacognitive sensitivity index meta-$d'$, a metric in the

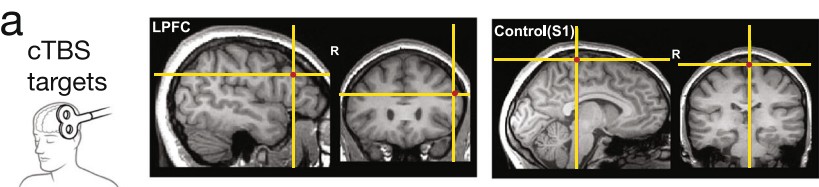

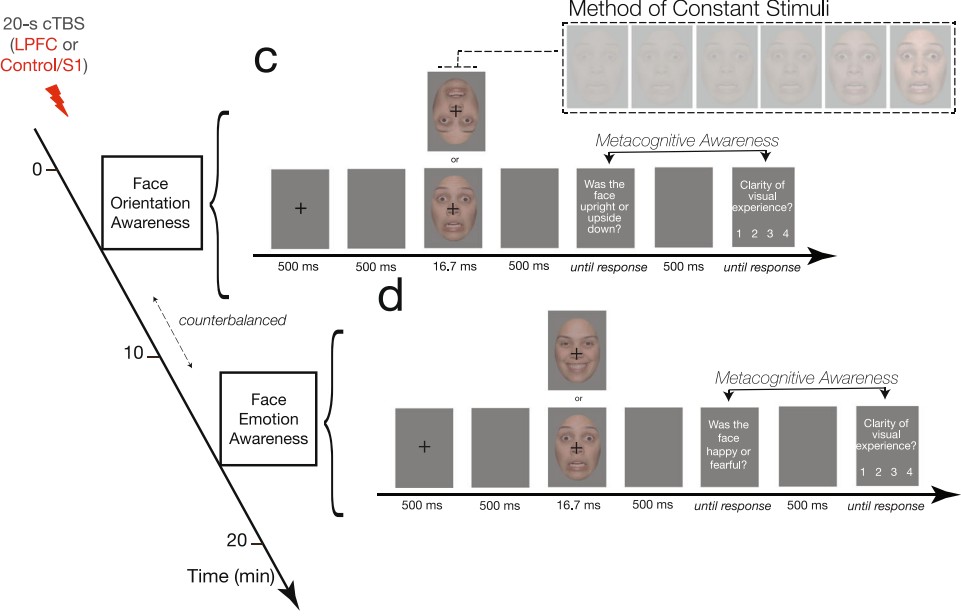

**Fig. 1 Experimental design of the TMS experiment (within subjects). a** LPFC and S1 (control) regions targeted in the administration of the continuous theta-burst transcranial magnetic stimulation (cTBS) protocol are shown, overlaid on a representative participant's T1-weighted image in native space. **b** Session procedures: Following 20-s (300-pulse) cTBS to LPFC or to S1 (order counterbalanced across subjects), participants performed two separate two-choice stimulus-discrimination tasks: one assessing metacognition of face orientation, and another of face emotion. Task order was counterbalanced across subjects. **c, d** The trial structure for the **c** orientation and **d** emotion discrimination tasks are shown. Faces were presented for 16.7 ms at six different contrasts using the method of constant stimuli (inset), after which participants were asked to perform a stimulus-discrimination judgment—face orientation (**c**) or face emotion (**d**) followed by a rating of their subjective visual experience using the Perceptual Awareness Scale[35]. Metacognitive awareness was assessed by quantifying the relationship between stimulus-discrimination accuracy and subjective visibility ratings (see Methods), where a higher correspondence between objective and subjective metrics of visual processing indicates higher metacognitive awareness.

same units as $d'$, which can then be used to estimate metacognitive capacity above and beyond task performance by directly comparing it to stimulus-discrimination performance ($d'$)—yielding an index of metacognitive efficiency. Following prior work examining prefrontal contributions to metacognitive efficiency[16,22], we examined the difference score: meta-$d'$ − $d'$. We found that cTBS applied to left mid-LPFC attenuates metacognitive awareness of face orientation (but not of face emotion) during near-threshold face processing. Collectively, these findings contribute to a prior controversial literature by demonstrating that (a) mid-LPFC function plays a causal role in metacognition, a role which (b) extends beyond simple visual stimuli to include introspective reports of naturalistic social stimuli (i.e., faces), while raising the possibility that (c) dissociations between modalities of metacognition (such as low-level visual vs. emotional) may occur during the processing of complex social stimuli.

## Results

**Overview**. In the following, we examined whether inhibitory cTBS to LPFC (vs Control/S1) modulated metacognition. We examined both metacognitive sensitivity (Type 2 AUC & meta-$d'$) and efficiency (meta-$d'$ − $d'$). Following prior work[16,21], we

examined metacognition at the contrast closest to participants' detection threshold (75%) using paired-samples $t$-tests. Next, using the full data obtained with the method of constant stimuli, we also probed whether inhibitory cTBS to LPFC altered metacognition independently of stimulus strength (i.e., contrast) using a repeated-measures analysis with cTBS (2) and stimulus contrast (6) as within-subjects factors. We first describe the results pertaining to the face orientation task given its direct relevance to prior work[16,21,30,31], followed by results pertaining to the face emotion identification task and a formal comparison between the two tasks.

**Face orientation metacognition**. By temporarily disrupting LPFC function using an inhibitory cTBS protocol, we impaired metacognitive awareness of the spatial orientation of emotional faces (Fig. 2). This effect was revealed by both metrics indexing metacognitive sensitivity during near-threshold processing, including the nonparametric Type 2 AUC $t(27) = −2.33$, $p = 0.027$, $d = 0.44$; as well as the Bayesian model-based meta-$d'$ $t(27) = −2.09$, $p = 0.046$, $d = 0.4$ (Fig. 2a, b). Metacognitive efficiency, which reflects metacognitive sensitivity relative to objective (i.e., stimulus detection) performance, trended in the same direction, meta-$d'$ − $d'$ $t(27) = −1.735$, $p = 0.094$, $d = 0.33$ (Fig. 2c).

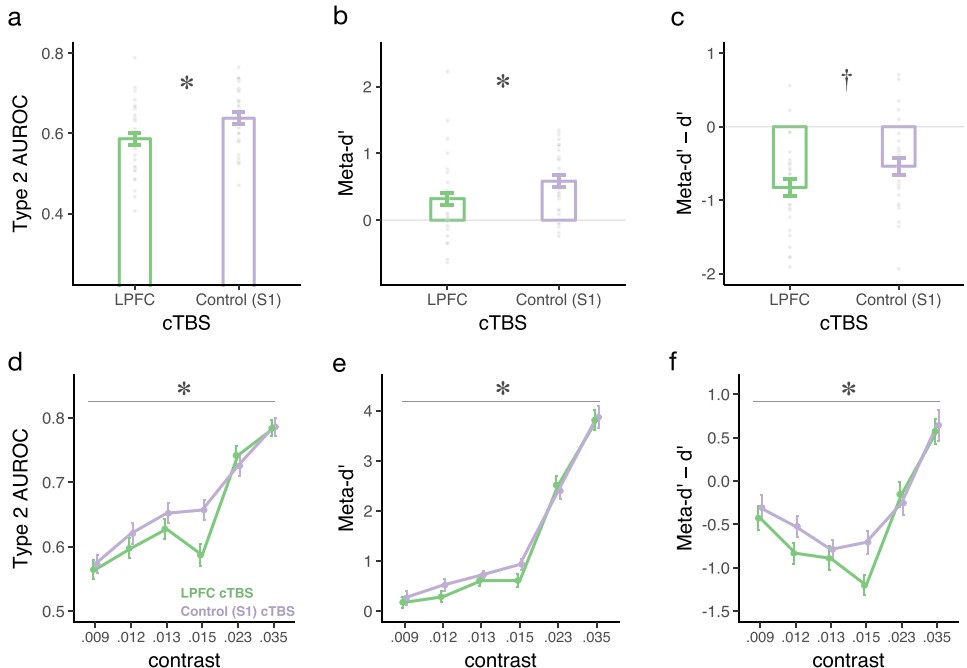

**Fig. 2 Metacognition of face orientation as a function of LPFC intactness.** Metacognitive sensitivity (**a**, **b**) and efficiency (**c**) of the orientation of emotional faces presented at the contrast nearest to each participant's threshold. Metacognitive sensitivity was significantly attenuated following inhibitory TMS (cTBS) to LPFC compared to a control site (S1) as indexed by both the nonparametric Area of Type 2 ROC metric (**a**) as well as the Bayesian model fit (meta-$d'$). **b** Metacognitive efficiency (i.e., metacognitive sensitivity, meta-$d'$, relative to objective stimulus-discrimination performance, $d'$) showed a similar trend. Each dot depicts the data of one participant ($N = 28$). Metacognitive sensitivity (**d**, **e**) and efficiency (**f**) of the spatial orientation of faces as a function of stimulus contrast (RMS) (i.e., independently of participant-specific thresholds). The impact of cTBS to LPFC (compared to S1) was reliably modulated by stimulus contrast (pronounced at intermediate contrast levels), as evidenced by significant contrast * cTBS site interactions across all three metrics of visual metacognition (denoted by the gray lines) ($N = 27$). Error bars represent within-subjects standard errors[60].

The method of constant stimuli allowed us to examine whether the impact of inhibitory TMS to LPFC on visual metacognition operated independently of the clarity of visual experience (i.e., across stimulus contrasts), or whether cTBS preferentially changed metacognition when perception was ambiguous (i.e., at intermediate stimulus contrasts). We found that stimulus contrast robustly modulated the impact of cTBS on metacognition of spatial orientation: Only at intermediate contrasts did LPFC cTBS attenuate metacognitive awareness of face orientation (Fig. 2d–f). This modulatory effect, evidenced by significant cTBS * contrast interactions, was present in all three metrics of metacognitive awareness: Type 2 AUC $W = 25.14$, $p = 0.007$, $\eta_p^2 = 0.492$; meta-$d'$ $W = 16.33$, $p = 0.043$, $\eta_p^2 = 0.386$; meta-$d' - d'$ $W = 26.83$, $p = 0.005$, $\eta_p^2 = 0.508$. The analysis of the association between visibility ratings and stimulus-discrimination performance at participants' near-threshold contrast indicated that subjective visibility following incorrect trials was rated higher following inhibitory cTBS to LPFC, thereby clarifying the nature of the reduced metacognitive awareness observed when LPFC function was altered. (Supplementary Fig. 1). Collectively, these results underscore that LPFC function causally promotes perceptual metacognition of complex human faces, particularly in ambiguous visual processing conditions.

Importantly, inhibitory cTBS to LPFC attenuated metacognition of face spatial orientation without impacting face orientation discrimination accuracy or overall subjective visibility, as evidenced by the nonsignificant impact of cTBS on $d'$ (cTBS main effect $p = 0.458$; cTBS * contrast interaction $p = 0.79$; Fig. 3a) and PAS (cTBS main effect $p = 0.14$; cTBS * contrast interaction $p = 0.36$; Fig. 3b), respectively. In sum, these results underscore the contribution of LPFC function for perceptual metacognition—i.e., one's ability to introspect into one's own

visual processing—as opposed to it impacting stimulus-discrimination performance ($d'$) or subjective visibility (PAS) per se.

**Face emotion metacognition.** In contrast with the above-reported findings, inhibitory cTBS to LPFC did not impair metacognition of face emotion (Fig. 4). This null finding was consistent across all metrics of metacognitive awareness, whether examined at near-threshold stimulus contrasts, Type 2 AUC $t(31) = 0.7$, $p = 0.49$, $d = 0.12$; meta-$d'$ $t(31) = 0.626$, $p = 0.54$, $d = 0.11$; meta-$d' - d'$ $t(31) = -1.15$, $p = 0.26$, $d = 0.2$ (Fig. 4a–c), or whether examined across all contrasts, Type 2 AUC $W = 7.08$, $p = 0.33$, $\eta_p^2 = 0.191$; meta-$d'$ $W = 4.728$, $p = 0.547$, $\eta_p^2 = 0.136$; meta-$d' - d'$ $W = 7.61$, $p = 0.286$, $\eta_p^2 = 0.202$ (Fig. 4d–f).

As with the face orientation task, face emotion discrimination performance ($d'$; cTBS main effect $p = 0.456$; cTBS * contrast interaction $p = 0.485$) and subjective visibility (PAS; cTBS main effect $p = 0.128$; cTBS * contrast interaction $p = 0.395$) remained unchanged following cTBS to LPFC (Fig. 3c, d).

**Comparing metacognition across face orientation and emotion domains.** Collectively, the results above suggest that LPFC computations contributing to metacognition may preferentially support metacognitive representations of low-level features such as orientation—i.e., perhaps relying on occipital-frontal projections, as opposed to putatively more distributed circuitry subserving the encoding of emotional valence (see Discussion). However, inferences about strong dissociations of metacognition due to LPFC function across the two face-discrimination tasks should be interpreted with caution, as the formal interaction of task by cTBS site reached significance only when examining

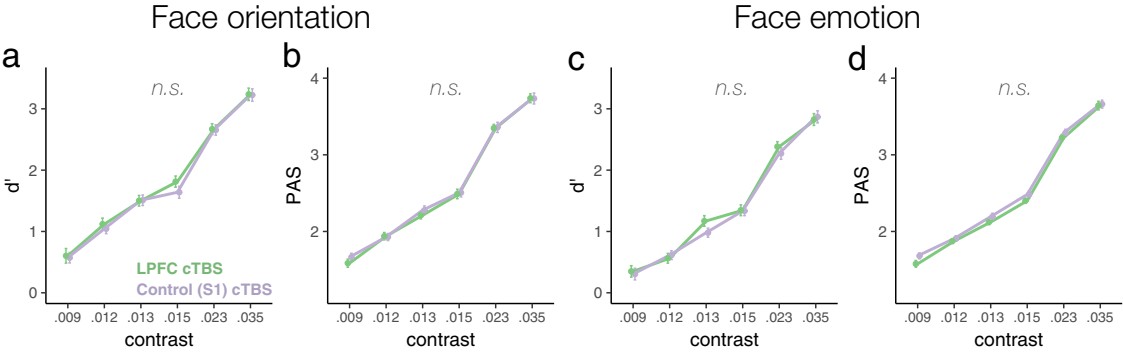

**Fig. 3 Stimulus discrimination accuracy and subjective visibility for human faces as a function of LPFC intactness.** Stimulus-discrimination accuracy (*d′*) and subjective visibility (PAS) are plotted as a function of cTBS and contrast (RMS) for spatial orientation (**a**, **b**) and emotion (**c**, **d**) face-discrimination tasks. cTBS to LPFC did not reliably change participants' accuracy (**a**) or subjective visibility (**b**) during face orientation discrimination, as indicated by the nonsignificant main effects of cTBS or cTBS * contrast interactions on *d′* or PAS (*N* = 27). Likewise, accuracy (**c**) and subjective visibility (**d**) remained unchanged by cTBS in the face emotion discrimination task (*N* = 31). Error bars represent within-subjects standard errors[60].

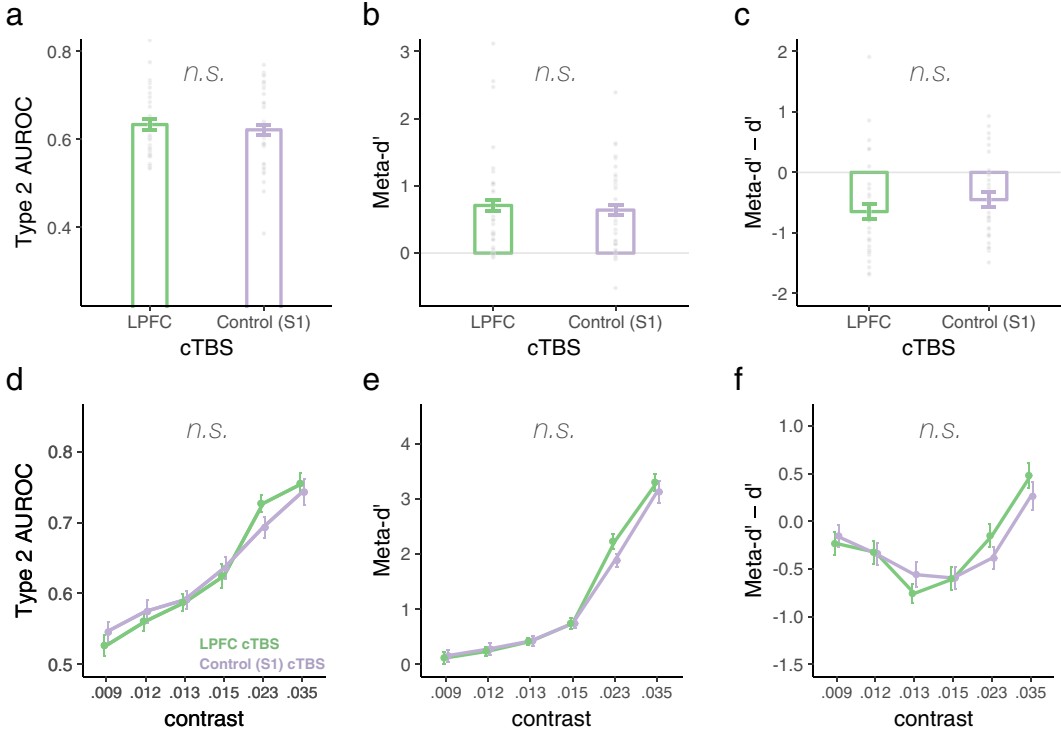

**Fig. 4 Metacognition of face emotion as a function of LPFC intactness.** Neither metacognitive sensitivity (**a**, **b**) nor efficiency (**c**) of emotion discrimination of emotional faces presented near-threshold were impacted by cTBS to LPFC. Each dot depicts the data of one participant (*N* = 32). Similarly, cTBS to LPFC did not impact metacognitive awareness of face emotion across contrasts, as suggested by nonsignificant cTBS main effects or cTBS* contrast interactions, *p*s > 0.26 (**d**-**f**) (*N* = 31). Error bars represent within-subjects standard errors[60].

metacognitive efficiency (i.e., controlling for task performance), meta-*d′* − *d′* (across contrasts), W = 5.183, *p* = 0.034 (Supplementary Results). Note that even though performance between the two tasks did not differ at participant' contrast threshold, *p* = 0.1, it did when the data were examined across all contrasts (Fig. 3a, c), W = 131.79, *p* < 0.001), thereby limiting the interpretability of metacognitive indices that do not control for task performance when examining the data across contrasts. Nonetheless, the consistently distinct pattern of results across face orientation and emotion discrimination tasks highlights the multifaceted nature of metacognition, and underscores the importance of multiple measurements in the same observers for a more nuanced understanding of the boundary conditions

underlying the contributions of prefrontal substrates to metacognition of complex stimuli.

## Discussion

How do we obtain accurate access to our ongoing internal experiences? Despite suggestive correlational studies, the causal contribution of prefrontal substrates to metacognition and conscious perception remained unsettled. Using a causal intervention (TMS), we demonstrate that function of the lateral prefrontal cortex promotes perceptual metacognitive awareness of one of the most common, salient, and motivationally informative stimuli that we encounter in everyday life—human faces. Of note, TMS

to LPFC impaired metacognition of face orientation while leaving unaltered metacognition of face emotion.

Although prior correlational evidence had suggested that function of anterior prefrontal (aPFC) and dorsal LPFC (DLPFC) promotes metacognition of low-level visual features[13,14,17,26,27,37], experiments using causal methods had provided mixed results, possibly due to in part to methodological inconsistencies. For instance, bilateral administration of cTBS to DLPFC reduced metacognitive awareness of geometric shapes[16]—but Bor et al. (2017) failed to replicate this effect, and Rahnev et al. (2016) (administering a longer cTBS protocol to right DLPFC) found evidence for a metacognition-enhancing (instead of impairing) effect[21,30]. However, Rahnev et al. (2016) targeted a DLPFC region anterior to the one targeted in the current study and in Rounis et al. (2010), thereby recapitulating metacognition-enhancing effects produced when the same group targeted an anterior PFC site (BA10)[30,31]. Moreover, neither Bor et al. (2017) nor Rounis et al. (2010) used MRI-guided TMS targeting. MRI-guided TMS has been shown to enhance (i.e., double) statistical power to detect TMS effects relative to external landmarks[38]. Therefore, it is possible that Bor et al.'s (2017) failure to replicate Rounis et al. (2010) may have been due to inadvertent stimulation of distinct locations within LPFC. Adding to this source of variability, the precise location of LPFC regions linked with metacognition has varied in prior work, and encompassed aPFC regions such as BA10[14,17] and rostrolateral BA46[14,27,29,39] as well as mid-lateral PFC 9/46v[26]; In Rounis et al. (2010), mid-LPFC sites targeted were estimated to fall within (left) 9/46v and (right) 8B[40].

Here, we used MRI-guided cTBS to inhibit function of a left mid-LPFC region in the inferior frontal sulcus, located between BA44 and 9/46 v—i.e., the estimated left-LPFC site targeted in Rounis et al (2010)[40,41]. Our results dovetailed with theirs by highlighting that inhibitory cTBS to LPFC impairs perceptual metacognitive awareness, even while leaving stimulus-discrimination performance (i.e., orientation judgments) intact.

Of note, two prior studies using MRI-guided TMS[30,31] concluded that the mid-LPFC (DLPFC) may not contribute directly to perceptual metacognition. In Shekhar & Rahnev (2018), online TMS to DLPFC lowered confidence ratings, without changing metacognitive awareness, whereas both offline and online TMS to aPFC have been found to increase metacognitive awareness (putatively by reducing metacognitive noise)[30,31]. In light of those data and a recent computational model, DLPFC was proposed to simply relay the strength of sensory evidence to aPFC (BA10), which would in turn transform the information received from DLPFC into a confidence judgment, and be the proximal site subserving metacognition[31]. Our findings are not incompatible with their proposal: if mid-DLPFC-originated sensory evidence precedes confidence computations in aPFC, local perturbations may ultimately cascade in altered metacognitive estimates computed by a later node. Nonetheless, it is important to note the neuroanatomical heterogeneity across DLPFC sites targeted in these studies. The left-DLPFC regions targeted here and in Rounis et al. (2010) are posterior and lateral to right DLPFC sites used in Shekhar & Rahnev (2018) & Rahnev et al. (2016). Accordingly, their functional connectivity fingerprints are distinct (Supplementary Fig. 3), with our site and Rounis et al.'s (2010) showing greater DLPFC-visual cortical coupling and a notably divergent profile of frontal network affiliation compared to Shekhar & Rahnev's (2018) and Rahnev et al.'s (2016); whether these distinct network affiliations account for the distinct functional contributions revealed by causal perturbations to those LPFC sites remains to be determined. Moving forward, dissecting the functional specialization of these separable prefrontal networks will likely prove critical for a thorough understanding of the neural architecture of metacognition.

In summary, our results extend the prior work on metacognition of spatial orientation[16,31] by highlighting that mid-LPFC causally promotes low-level perceptual metacognition not only of simple nonsocial stimuli, but also of complex, naturalistic human face stimuli, which comprise an essential source of motivationally relevant information that guides our adaptive behavior in a multitude of contexts. Collectively, these results clarify the import of LPFC function to visual metacognition in naturalistic settings and reinforce a role for prefrontal substrates in conscious visual perception[19,20].

Do LPFC computations that support metacognition represent actual contents of conscious experience, or do they instead enable conscious access to sensory representations stored elsewhere? Current accounts (e.g., ref.[31]) largely support the latter viewpoint, wherein LPFC reads out the strength of sensory information from lower-order cortices, which is consistent with the extensive connectivity between frontoparietal and multimodal temporal cortex[42]. According to that view, LPFC's role provides a background condition for accurate introspective access, or visual consciousness[20]. However, it is possible that LPFC representations give rise to a unique type of conscious content—the "feeling of knowing" that one is perceiving something. Accordingly, domain general and specific reports of confidence about perceptual experiences have been decoded from multivariate patterns in LPFC[13] and used to specifically manipulate confidence without altering perceptual discrimination performance[43].

Is LPFC function in metacognition domain-specific? In this study, we probed metacognition of faces using an approach that is well aligned with the extant literature on perceptual (visual) metacognition, which has often adopted stimulus types and discrimination tasks in which orientation was a core discerning feature[16,31]. In order to glean insight into the domain generality of lateral prefrontal contributions to metacognition of complex social stimuli, we also separately examined metacognition of face emotion, a core feature of human faces[12,44]. Our data showed preliminary evidence for a possible dissociation between metacognitive judgments for orientation vs. emotion features of complex face stimuli. In contrast with robust attenuation of metacognition of face spatial orientation following cTBS to LPFC, metacognitive awareness of emotional expressions was largely unaffected by cTBS, suggesting a possible dissociation of the neural substrates supporting metacognition of these two important face features. However, as the test of interaction between task and cTBS site only reached significance for a metric of metacognitive efficiency (meta-$d'-d'$) and not for metacognitive sensitivity alone (AUC and meta-$d'$), we are cautious in interpreting this effect, and hope that it paves the way for future studies. For instance, it is possible that metacognition of emotional expressions relies on re-representations of emotional valence in superior temporal and medial frontal (including interoceptive) circuitry, rather than on occipital-LPFC projections[45]. Consistent with this idea, a recent study found that metacognitive awareness of emotional expressions correlated with function and white matter microstructure of the cingulate cortex, and not of LPFC[24]. As a rigorous neuroscience of emotional consciousness is in its nascent stages[8,33], carefully delineating first and higher-order correlates of human affective encoding and experiences—and testing their causal contribution to conscious emotional states—will be critical avenues for future work.

Metacognitive efficiency is thought to index metacognition above and beyond task performance (i.e., by subtracting stimulus-discrimination performance ($d'$) from metacognitive sensitivity measured in the same unit (meta-$d'$)[46]. Nonetheless, in our study, metacognitive efficiency (meta-$d' - d'$) varied across face contrast (Figs. 2f and 4f), exhibiting a U-shaped curve in which lowest and highest contrasts produced higher metacognitive efficiency than

intermediate contrasts. Recent computational models suggest two mechanisms that may account for this pattern: (a) Changes in sensory noise at low/intermediate contrasts: a recent hierarchical model of confidence predicts that sensory noise produces higher estimates of metacognitive efficiency[47]. According to this model, metacognition (meta-$d'$) is corrupted by both sensory and metacognitive noise, whereas stimulus-discrimination performance ($d'$) is corrupted by sensory noise only—thereby giving rise to non-constancy in metacognitive efficiency (meta-$d' - d'$) across different sensory noise levels. Perceptual learning, which reduces sensory noise, has been shown to (perhaps counterintuitively) reduce metacognitive efficiency (meta-$d' - d'$)[47]. It is possible that sensory noise varied by contrast in our study. For instance, if some degree of learning took place for near-threshold stimuli over the course of the TMS session, reduced sensory noise would account for lower metacognitive efficiency estimates at intermediate (but not lowest) contrasts. (b) Error detection at high contrasts: Theoretically, meta-$d' - d'$ should be as high as 1 (i.e., perfect metacognition)—but in practice, "hyper metacognitive sensitivity" (meta-$d' - d' > 1$) has been often observed empirically (Figure 8E in ref. [48]). This phenomenon can be explained in part by post-decisional factors, such as error detection. For instance, as stimulus strength increases (e.g., at high contrasts), errors are less frequent and more obvious, and more likely to be driven by motoric or attentional lapses (as opposed to sensory noise). The introspective observer realizes potential motoric and attentional lapses when the sensory information is clear, resulting in greater error detection. Accordingly, a Bayesian model of confidence that incorporates post-decisional factors accounts for this phenomenon: reliable error detection gives rise to hyper metacognitive sensitivity (i.e. meta-$d' - d' > 1$)[48].

The following limitations of the current study warrant additional investigation. First, we targeted left-LPFC based on a neuroimaging experiment examining the neural correlates of emotional-face awareness[49], which agreed with prior lesion and neuroimaging evidence pointing to neural correlates of subjective visibility in the left-LPFC[29,39]. However, stronger right-hemispheric involvement in face perception has been documented[50,51]—therefore, it is possible that the impact of cTBS to LPFC on face metacognition would have been greater, or would have extended to the domain of face emotion, had cTBS been administered to right LPFC. Additionally, we used an MRI-guided cTBS approach to ensure neuroanatomically consistent targeting. However, the lateral prefrontal wall is amongst the most recently developed and heterogeneous regions of the frontal lobe, such that idiosyncratic patches with distinct network affiliations may be present across individuals in seemingly anatomically consistent sites[52,53]. Thus, incorporating individualized and network-based LPFC parcellation strategies in future work may bring unique insights into the functional organization of metacognition.

In closing, our results indicate that LPFC function supports perceptual metacognition for a ubiquitous class of complex and naturalistic stimuli: human faces. Moving forward, it will be critical to understand how metacognition of their distinct social and emotional features are organized to inform optimal decision making, build cohesive subjective experiences, and permit our accurate understanding of others.

## Methods

**Power analysis.** The sample size for the present study was determined based on a power analysis performed on data from the most-pertinent published experiment probing the causal role of LPFC function on visual metacognition via cTBS that was available at the time of participant recruitment[16]. In that study, statistical power obtained for the paired-mean difference of metacognitive awareness sensitivity following cTBS to LPFC vs. sham was $d = 0.693$, which required a sample size of $n = 19$ to detect a statistically significant effect at alpha two-tailed $p < 0.05$ and power = 80%.

**Participants.** Therefore, with the goal of retaining a minimum of $n = 19$ participants with useable data across the multiple ($n = 2$) TMS sessions and ($n = 4$) psychophysical assessments, we recruited 34 right-handed individuals from the University of Wisconsin–Madison. One participant did not tolerate TMS delivered to the LPFC target, therefore rendering the maximum sample size $N = 33$ (19 males; age 18–32; $M = 23.79$, $SD = 4.428$). As detailed below, $n = 28$ participants (16 males; age 18–32; $M = 23.5$, $SD = 4.718$) provided useable data for the task assessing metacognition of face orientation, and $n = 32$ (19 males; age 18–32; $M = 23.81$, $SD = 4.497$) provided useable data for the task probing metacognition of emotional expressions. Prior to participating, participants were screened in a clinical interview for neurological and psychiatric conditions, as well as for TMS and MRI safety criteria. The study protocol was approved by the University of Wisconsin–Madison Health Sciences Institutional Review Board. All subjects provided written informed consent.

**Procedure.** Overview: Following clinical screening, participants underwent an MRI session where T1-weighted scans were obtained to enable subject-specific neuro-navigation and accurate TMS targeting. On a separate day, the TMS session took place, wherein continuous theta-burst TMS (cTBS) was delivered to both left-LPFC and to a control site (left S1) within-subjects, with TMS site order counterbalanced across participants (Fig. 1a, b). Participants then completed two separate stimulus awareness tasks, one indexing metacognition of face orientation, and another indexing metacognition of face emotion (Fig. 1c,d). In between TMS administrations to LPFC and control sites, participants took a 15-min break.

The experimental design is within-subjects. Assignment to "LPFC site first" vs. "S1 site first" was based on subject number (odd vs. even). Assignment of face orientation and face emotion task order was independent of TMS site order (awareness task order alternated every $n = 4$ sessions). During the TMS session, a minimum of $n = 2$ researchers were always present (RCL and a research assistant).

MRI session and acquisition parameters: MRI data were acquired with a 3.0 T GE scanner (GE Healthcare, Waukesha, WI) using an 8-channel coil. High-resolution 3D T1-weighted inversion recovery fast gradient echo (Mugler, 1990) anatomical images were collected in 160 contiguous 1.25 × 1.25 × 1.25-mm sagittal slices (TE = 2.3 ms; TR = 5.6 ms; flip angle = 12°; FOV = 240 × 240 mm; 192 × 192 × 160 data acquisition matrix, inversion time TI = 450 ms).

TMS site: The left mid-LPFC site targeted in this study (Fig. 1a) is located at MNI coordinates [x, y, z] = [−48, 24, 20]. This LPFC site was chosen based on prior fMRI work examining awareness-dependent changes in neural circuitry underlying negative-face processing[49], wherein BOLD activation in this region was found to be significantly increased by visual awareness, and its functional connectivity with the amygdala associated with differential behavioral regulatory-outcomes as a function of visual awareness. The targeted mid-LPFC region is located near the inferior frontal sulcus, and estimated to be between BA44 and BA9/46 v[40,41], regions highly interconnected with the frontoparietal network and multimodal temporal cortex[42].

In order to identify the LPFC site for TMS targeting on a subject-by-subject basis, a 12-df affine registration was performed between each participant's T1-weighted scan and the MNI template. Then, the registration matrix was inverted, and the LPFC target was registered to each participant's native space. Next, each participant's native space target was visually inspected to ensure satisfactory registration and target placement on grey matter.

As a control TMS region, we targeted the left medial somatosensory cortex (S1), in a region consistent with the sensory representation of the right foot (approximate MNI coordinate [−10, −38, 78], thereby avoiding inadvertently stimulating lateral, face-representation areas; Fig. 1a). The S1 target was located on each subject's native space T1-weighted image based on anatomy. This region was chosen as an active TMS control region due to its circumscribed functional connectivity, and because this approach permits us to rigorously control for non-specific effects of stimulation of brain tissue[54].

TMS stimulation protocol: TMS was delivered to the left-LPFC and to medial S1 with a Magstim Super Rapid magnetic stimulator (Magstim, Whitland, UK) equipped with a figure-8 stimulating coil. Precise TMS targeting on a subject-by-subject basis was achieved via a Navigated Brain Stimulation (NBS) system (Nextstim, Helsinki, Finland), which uses infrared-based frameless stereotaxy to map the position of the coil and the subject's head in relation to the space of the individual's high-resolution MRI.

In order to temporarily interfere with function of LPFC and Control/S1 sites, we used a continuous TMS protocol—cTBS—consisting of 50 Hz trains of 3 TMS pulses repeated every 200 ms continuously over a period of 20 s (300 pulses total). This 20-s cTBS protocol has been shown to depress activity in the stimulated brain region for up to 20 min after stimulation[55].

As is typical with this TMS protocol, we delivered cTBS at 80% of active motor threshold. The active motor threshold was defined as the lowest stimulus intensity that elicited at least five twitches and/or sensations in 10 consecutive stimuli delivered at the motor cortex while the subject maintained a voluntary contraction of index and thumb fingers at about 20% of maximum strength. cTBS was delivered with the coil placed tangentially to the scalp, and with the handle pointing posteriorly. The stimulation varied between 32 and 57% of the maximum stimulator output (0.93 T at coil surface) ($M = 51.18\%$, $SD = 6.23\%$).

TMS session procedures: The TMS session began with a broad overview of the experiment. Participants sat at a chair with their eyes positioned 80 cm away from a

computer monitor (ASUS HDMI set to 60 Hz refresh rate; 53 cm screen width; 1920 × 1080 pixels resolution). Then, they were introduced to the Perceptual Awareness Scale (PAS) and practiced the tasks. As part of a larger study, they underwent sensor placement for EEG recordings (described in ref. [56]).

Next, cTBS was administered for 20 s to LPFC or to the Control/S1 site, with TMS site order counterbalanced across participants. Participants completed two separate face-stimulus-discrimination tasks, one indexing metacognition of face orientation, and another indexing metacognition of face emotion, in counterbalanced order (Fig. 1b). As part of a larger study, participants underwent another cTBS administration to each cortical site followed by an experiment assessing emotion regulation (as reported in ref. [56]); experiment order (metacognitive awareness vs. regulation first) was counterbalanced across subjects.

In the middle of the experiment (i.e., following TMS administration to LPFC or Control/S1) participants took a 15-min break. Then, the sequence of steps delineated above was repeated, with cTBS administered to the other site (LPFC or Control/S1). The full experiment was run using PsychoPy 2 (v. 1.79.01)[57].

Stimuli: Emotional faces (happy and fearful) consisted of 24 identities (half female) selected from the Karolinska Institute Set http://www.emotionlab.se/resources/kdef and the Macbrain Face Stimulus Set http://www.macbrain.org/resources.htm). We matched both average luminance and RMS contrast across faces. Faces were cropped to remove hair and neck. Two stimulus sets comprising 12 identities each were created and assigned to LPFC and control TMS conditions in a counterbalanced manner across subjects. Emotional faces were presented at 6° × 6° using PsychoPy[57]. The full list of emotional-face stimuli, example stimuli, and stimulus presentation scripts used in this study are available online: https://osf.io/t8m4j/.

Face-discrimination tasks: Metacognitive awareness of emotional faces was assessed using two separate tasks: one in which participants discriminated the orientation that emotional faces (upright or upside down), and another in which they discriminated their emotional expression (fearful or happy) (Fig. 1c, d). Stimulus-discrimination responses were followed by an assessment of participants' subjective visual experiences as detailed below, thereby providing the data required to compute metacognitive awareness—i.e., the extent to which participants' subjective visibility ratings tracked their objective stimulus-discrimination performance.

We used the method of constant stimuli in the face orientation and emotion discrimination tasks. Each face was presented at six different contrasts (RMS: 0.009, 0.012, 0.013, 0.015, 0.023, 0.035). This approach was chosen for two reasons: First, it allows for the estimation of metacognitive awareness around participants' discrimination threshold (75% accuracy)— thereby permitting more direct comparisons with prior work[16,21]. Second, this approach also allows us to probe the contributions of LPFC to metacognitive awareness across a wider range of visual experiences than previously assessed, including sub- and super-threshold levels. The contrasts adopted in our study were chosen empirically, initially based on the behavioral performance of three of the authors (RCL, BR, JS), followed by iterative refinement based on behavioral piloting using a total of $n = 13$ naive observers (undergraduate students at the University of Wisconsin–Madison). Our goal was to find a set of contrasts that reliably captured the psychometric curve of most or all observers, including performance near-threshold (75%). The set of contrasts used here was tested on three of the authors (RCL, BR, JS) and $n = 9$ naïve observers, and captured threshold performance of $n = 11/12$ of them.

Each discrimination task had a total of $n = 144$ trials (twenty-four emotional faces presented per contrast). Emotional-face-identity (and its two facial expressions) was randomly assigned to one contrast. Emotional face contrast was altered using the opacity parameter in PsychoPy (opacities: 0.08125, 0.10, 0.113, 0.127, 0.198, 0.30). Emotional faces were presented upright in the emotional-expression discrimination task, and half upright, half upside down in the orientation discrimination task. Each orientation- and emotional-expression-task took ~10 min to complete, totaling ~20 min for both tasks.

The trial structure is detailed in Fig. 1c, d: Emotional facial expressions were shown for 16.7 ms. Individuals were first asked to discriminate the stimulus (Orientation task: "Was the face upright or upside down?"; Emotion task: "Was the face happy or fearful?"). Next, participants indicated the clarity of and confidence in their visual experience using the Perceptual Awareness Scale; PAS[35], where "1" = "No experience (you could not see a face, and guessed your answer)"; "2" = "Brief glimpse (you have a feeling that a face might have been shown, but you cannot indicate its orientation/expression)"; "3" = "Almost clear experience (Ambiguous visual experience of the face; you are almost certain of your answer)"; "4" = "Clear experience (Non-ambiguous visual experience of the face; you are certain of your answer)". Finally, a brief inter-trial interval (1–1.5 s, sampled from a uniform distribution) followed.

Metacognitive awareness estimation: Metacognitive visual awareness refers to one's subjective access to their visual experiences, and is computed by estimating how well subjective visibility ratings distinguish between correct and incorrect responses—in other words, do participants' subjective visual experiences (indexed by the PAS[35]) mirror their actual performance (i.e., stimulus-discrimination accuracy)? For completeness and comparability with prior relevant TMS work[16,21], we estimate visual metacognition per subject, cTBS condition, and contrast using three distinct methods frequently employed in the literature: Type 2 AUC, meta-$d'$, and meta-$d' − d'$. We briefly review these methods below, as they have been covered in detail elsewhere[25].

Type 2 AUC is a robust nonparametric method for estimating metacognitive sensitivity. It entails first computing Type 2 "hit rates", defined as p(Confidence |

Correct trials), as well as Type 2 "false alarms", p(Confidence | Incorrect trials). At multiple levels of confidence as is the case with the PAS, the full Type 2 ROC can be constructed by successively treating each confidence rating as a criterion that separates "high" from "low" confidence[25]. The area under the constructed Type 2 ROC is what we term Type 2 AUC. When the Type 2 AUC is large, a participant's subjective ratings closely tracks their discrimination performance, and is said to have high metacognitive sensitivity. This method has the advantage of being robust to normality (Gaussian) assumptions and modest trial counts. We used the Type 2 ROC code provided here: https://github.com/metacoglab/meta_dots/blob/master/type2roc.m.

A more recently developed Bayesian model-based method, meta-$d'$, takes a different approach to estimate metacognitive sensitivity: Given the subjective visibility ratings (PAS) reported by an observer, and assuming Gaussian distributions at the stimulus-discrimination level, one can estimate the stimulus-discrimination accuracy ($d'$) most likely to have given rise to the data in a metacognitively ideal observer—this is meta-$d'$. This metric, meta-$d'$, is expressed in the same signal-to-noise ratio as $d'$, and therefore permits a direct comparison between performance and metacognition (i.e., metacognitive efficiency), as detailed next. To compute meta-$d'$, we used the single-subject Bayesian meta-$d'$ algorithm, which does not use zero-cell count correction and thus is robust to low trial numbers relative to earlier meta-$d'$ implementations (https://github.com/metacoglab/HMeta-d/blob/master/Matlab/fit_meta_d_mcmc.m[36]). At participants' near-threshold contrast, zero counts did not differ between TMS conditions in either the face orientation task, Fisher's $p = 0.66$, or in the face emotion task, Fisher's $p = 0.92$ (Supplementary Tables 1 and 2). Zero counts also did not differ when collapsed across contrasts, orientation task χ2 = 0.92, $p = 0.996$; emotion task χ2 = 2.629, $p = 0.917$.

Metacognitive efficiency (as estimated by the difference score meta-$d' − d'$) was the primary metric used in the TMS work that motivated the current study[16,21]; we therefore report it here. Meta-$d' − d'$ has an intuitive interpretation, as it indexes participant's metacognitive sensitivity while adjusting for the influence of task performance. For example, if the observer is metacognitively ideal (i.e., she is fully aware of the sensory information that informed the (objective) discrimination performance), then meta-$d' − d' = 0$. If she is metacognitively suboptimal, meta-$d' − d' < 0$.

**Statistics and reproducibility**. Participant inclusion criteria and statistical power: We used the method of constant stimuli to determine psychophysical performance and estimate metacognition for near-threshold stimuli (as per prior work[16,21]), as well as to examine the potential role of LPFC in promoting metacognition outside of the near-threshold range. Estimated stimulus-detection thresholds fell within the contrast range spanned by the face stimuli (i.e., where the 95% confidence interval of objective stimulus-discrimination performance included 75% accuracy for at least one contrast) for 28 (out of 33) participants in face orientation discrimination task ($n = 5$ participants were at ceiling), and for 32 (out of 33) participants in the emotional-expression discrimination task ($n = 1$ participant was at floor)—those participant groups therefore comprise the final sample used for data analysis.

Based on the aforementioned power analysis and final sample sizes, we estimate that obtained power = 94.2% to detect a significant reduction of metacognitive awareness due to cTBS in the face orientation task, and power = 96.7% to detect a cTBS induced change in metacognition in the face emotion task (at alpha < 0.05). Results with all participants included (regardless of whether their near-threshold performance was captured) are reported as Supplementary Results.

Data reduction and analysis: All statistical analyses were conducted in R version 3.4.1, using R Studio Version 1.0.153 and the following packages: tidyr_0.7.2, dplyr_0.7.4, reshape2_1.4.2, ggplot2_2.2.1, binom_1.1-1 and MANOVA.RM_0.3.2. All statistical tests were two-tailed.

Using the data obtained through the method of constant stimuli, we identified the contrast level closest to threshold performance (75% accuracy) for each participant and task. Metacognitive awareness assessed at the near-threshold contrast was examined as the primary outcome measure for best comparability with prior work, which had estimated metacognition at participants' threshold performance (75%) using staircase procedures. As mentioned in the Results, cTBS did not reliably impact stimulus-discrimination performance ($d'$) in either the orientation or emotion task when examined across contrasts ($p$s > 0.4), nor did it significantly change the contrast which was closest to participants' threshold performance ($p$s > 0.29)—therefore, we estimated participants' near-threshold face contrast collapsing across cTBS conditions as to increase threshold-estimation reliability. Note, however, that when examining performance at the near-threshold contrast in the face emotion task, a modest impact of cTBS was observed on $d'$, $t(31) = 2.12$, $p = 0.043$, $d = 0.37$ (accounted for in metacognitive efficiency estimate meta-$d' − d'$). Near-threshold performance ($d'$) in the face orientation task was not impacted by cTBS, $t(27) = 0.24$, $p = 0.81$, $d = 0.04$ (Supplementary Fig. 2).

Metacognition estimates at the near-threshold contrast following cTBS to LPFC vs. Control/S1 (Type 2 AUC, meta-$d'$ and meta-$d' − d'$) were compared using paired-samples t-tests. We hypothesized that cTBS to LPFC would attenuate metacognitive sensitivity and efficiency (relative to the Control/S1 cTBS condition).

As a secondary question, the data obtained through the method of constant stimuli allowed us to explore whether the putative causal role of LPFC function in promoting visual metacognition would manifest primarily in situations of perceptual ambiguity (i.e., near-threshold, as previously demonstrated), or

whether it would be present across a wider range of participants' visual experiences (i.e., independently of stimulus contrast). To answer this question, we entered metacognitive awareness scores obtained for each cTBS condition and stimulus contrast in a repeated-measures analysis with cTBS (2) and stimulus contrast (6) as within-subjects factors. This allowed us to test whether the impact of cTBS depended on contrast, formalized as a significant cTBS * contrast interaction. (Note that the first participant run in this experiment had a different set of contrasts than the remaining (RMS: 0, 0.009, 0.012, 0.015, 0.035, 0.113). As a result, they are included in the primary (threshold-based) analysis as we were able to capture their performance threshold, but not in the secondary (contrast-based) analysis due to non-equivalent contrast sets.)

We estimated the cTBS * contrast interaction using the Wald Type Statistic (W), a method which is robust to violations of the assumptions of sphericity, compound symmetry, and multivariate normality, which are problematic (and often violated) when conducting repeated-measures ANOVAs on within-subjects factors that exceed two levels (in our case, $n = 6$ levels for stimulus contrast)[58]. We report the significance level of W obtained using the permutation-based resampling of the WTS, which has been demonstrated to be superior (i.e., more stable and with smaller Type I errors) relative to the asymptotic $\chi^2$ distribution or the ANOVA bootstrap-based approximation when sample sizes are small to moderate[59].

**Reporting summary**. Further information on research design is available in the Nature Research Reporting Summary linked to this article.

## Data availability

Experimental materials (example stimuli and code written in PsychoPy) are publicly available via Open Science Framework and can be accessed at https://osf.io/t8m4j/. Institutional Review Board constraints at the University of Wisconsin–Madison Health Sciences IRB precluded the authors from publicly sharing the raw data. The raw data are being stored on a secure server at the University of Wisconsin–Madison and may be available upon reasonable request from the corresponding author contingent on IRB approval. All source data underlying the figures in the manuscript (Figs. 2a–f, 3a–d, 4a–f, Supplementary Figs. 1 and 2a–d) are available via the Open Science Framework and can be accessed at: https://osf.io/dgmz9/.

## Code availability

The code for statistical analysis and figure generation was written in R and it is publicly available via the Open Science Framework: https://osf.io/xd7wq/.

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

## Acknowledgements
This study was supported by National Institute of Mental Health Grants MH113347 (R.C. Lapate), MH43454, MH069315 (R.J.D.) and MH095984 (B.R.P.). We would like to thank Mark D'Esposito for mentorship and support, Ali Austermuehle and Mike Starrett for administrative assistance, Dan Grupe, Robin Goldman, Olivia Gosseries, Andy DeClercq, Helen Weng, and Yuan Chang Leong for discussions, and the staff in the Waisman Laboratory for Brain Imaging and Behavior for technical support. Development of the MacBrain Face Stimulus Set was overseen by Nim Tottenham and supported by the John D. and Catherine T. MacArthur Foundation Research Network on Early Experience and Brain Development. Please contact Nim Tottenham at tott0006@tc.umn.edu for more information concerning that stimulus set.

## Author contributions
R.C.L. and R.J.D. developed the study concept. B.R., B.R.P., J.S., R.C.L., and R.J.D. contributed to the study design. R.C.L. performed testing and data collection. R.C.L. analyzed the data; J.S. contributed analytical tools. R.C.L. wrote the first draft of the manuscript. B.R., B.R.P., J.S., R.C.L., and R.J.D. contributed to and approved of the final version of the manuscript.

## Competing interests
The authors declare no competing non-financial interests but the following competing financial interests: R.J.D. is the founder, president, and serves on the board of directors for the non-profit organization, Healthy Minds Innovations, Inc. In addition, R.J.D. served on the board of directors for the Mind & Life Institute from 1992–2017. The remaining authors declare no competing financial interests.
