## [Peer Review File · Communications Biology]

Reviewers' comments:

Reviewer #1 (Remarks to the Author):

In this study, Lapate et al., investigate the causal effect of TMS to LPFC on metacognitive functioning. Specifically, they find that interfering with LPFC activity impairs metacognitive awareness for the orientation (but not emotion) of human faces. I appreciate their completely within-subjects study design which allows for clean comparisons between different experimental conditions and their efforts to collect a relatively large sample of participants. Their results also potentially suggest a functional dissociation between the neural substrates which support metacognition for low-level stimulus features and those which support metacognition for emotion. I think that this dissociation is very interesting and can further inform the domain specificity vs generality debate on metacognition. Their finding of an interaction between TMS condition and contrast level for the face orientation task is also quite clear and convincing. Overall, their study contributes to existing evidence on the causal involvement of PFC in perceptual metacognition and extends these findings to perception of complex and ecologically relevant stimuli such as human faces.

My main concerns are related to the stability of their results, due to low trial counts. I think if the authors can demonstrate the robustness of their findings to alternative methods of analysis, their overall results can be considered more reliable and convincing. I outline my concerns in more detail below:

1. In their methods, they mention that the six presented levels of stimulus contrast consisted of 24 trials each. Given that subjects performed at an accuracy of 75% near their threshold, this would only give them 6 error trials on average, which are critical for computing metacognitive measures such as Type-2 AUC and meta- d' (which essentially rely on estimates of false alarms). Due to this issue, I am seriously concerned about how reliable their estimates of metacognitive ability are and whether their effects are being biased by applying corrections to the zero cell counts (for example, for incorrect trials with a visibility of 4). In this regard, can they report whether the observation of zero cell counts remained the same or varied between the two TMS conditions?

I can also think of a couple of ways in which the authors can support their findings by demonstrating the reliability of their results. One might be to increase the trial counts for each cell by doing a median split on the confidence scale. In this way, some of the zero-count issues for extreme cells can be avoided. Although using a 4-point scale certainly affords better resolution of confidence, given that trial number can be problematic, I think that in this case, such an analysis might be useful to compound their results. Second, I would suggest pooling contrasts which are close to (or flanking) the threshold for each subject to increase their trial counts. Again, I am aware of the potential disadvantage of pooling contrasts (Rahnev and Fleming (2019) have demonstrated that doing this can inflate estimates of metacognitive ability). However, given that all the subjects experienced the same set of contrasts for all conditions, pooling nearby contrast levels should not confound their results.

2. In Figure 2, they report comparisons of metacognitive measures between the two TMS sites, analyzed only for stimuli at the contrast threshold. The difference between metacognitive efficiency (meta- $d'-d'$) for the two conditions is not significant, which makes me wonder whether some of the changes in meta- d' are being driven by changes in d' . Although they do report d' comparisons separately for the 6 contrast levels, I think it is more important to show these comparisons for the threshold stimuli, because it is for these stimuli that they observe the strongest effect on metacognitive awareness. Relatedly, since the LPFC they targeted also covers the dorsolateral prefrontal cortex (BA46), I am curious whether they observed any changes in mean visibility (since previous studies have shown an effect of DLPFC stimulation of mean confidence/visibility as well). If

this effect is absent, could the authors comment on why they think this difference may be arising?

3. Lastly, in the introduction they mention a number of motivating factors for their study such as the controversy surrounding the causal status of PFC in metacognition, the directionality of its functioning and the debate on whether metacognition is domain general or specific. However, I feel that their discussion of their findings does not fully address how these debates or questions are being informed by their current study. For example, they discuss that the controversy surrounding the findings of Rounis et al (2010) may be attributed to their lack of MRI guided neuronavigation. However, they also cite other TMS studies which have used such anatomically/functionally guided neuronavigation techniques for application of TMS (Rahnev et al., 2016; Shekhar and Rahnev, 2018). So, if they do want to address this controversy in their discussion, I think it would be important to include it within the context of all these existing studies and specify how their study can additionally contribute to resolving the debate.

Additionally, it may be helpful to include subsections in the discussion for each of these three motivating questions and specifically address how their findings can contribute to our understanding in these areas. Also, at the very end of the introduction, where the authors report their findings, I think it would also be useful to briefly mention the implications of their findings within the context of their motivation.

Minor comments:

1. I think the domain generality vs domain specificity effect in the introduction (p4, line 7) can be expanded a little more for context and clarity, if I am correctly assuming that this is also one of their main research objectives.
2. At the beginning of the results section, a brief summary of the methods (task and comparisons) and objective of the study will be useful to anchor the readers.
3. In the reporting of results on p10, why are the actual p-values not being reported for the analyses on the near threshold stimulus contrasts? I think there should only be a single p-value for paired t-tests in these analyses.
4. Their results suggest an interpretation that PFC is involved in perception of low-level features of faces but not their emotional content. However, I am not very convinced by the emphasis on 'orientation of emotional faces' (abstract, line 11) because their study design does not support comparisons of emotional faces against neutral faces. More specifically, without knowing how their results might have differed for orientation discrimination on neutral faces, the emphasis on emotional faces does not seem very meaningful here.
5. Can the authors add some details explaining how they decided on the 6 different contrast levels used for the experiment?
6. Metacognitive efficiency, measured by meta- $d'-d'$, is thought control for the effects of variation in task performance. So, I wonder why there is so much variation in meta- $d'-d'$ across the 6 different contrast conditions. Can the authors comment on this?

References:

Rahnev D, Fleming SM (2019) How experimental procedures influence estimates of metacognitive ability. *Neurosci Conscious* 2019.

Reviewer #2 (Remarks to the Author):

Manuscript#: COMMSBIO-19-1456-T

Summary: The study aims to investigate a possible causal link between LPFC activity and metacognitive ability, as suggested previous work in the literature. By administering MRI navigated cTBS protocol with a paradigm that varies the noise in the visual stimuli, the study demonstrates that LPFC supports metacognitive ability in perceptual metacognition of face orientation, but not of emotional expression of faces.

Overall Impression: I much enjoyed reading this manuscript. It's a very well written paper on a study that employed very clean manipulations bringing together different methodologies to address a very specific question. It's a novel and valuable contribution to the literature. My own work will certainly benefit from it.

Comments on Introduction and discussion:

1. Authors would want to incorporate a discussion in the intro (and perhaps in the discussion as well) to explain the link between consciousness and metacognition, and what's exactly referred to with 'metacognitive awareness'. Metacognition can also operate implicitly, and the prompt used in this paradigm ('how clear was your experience') asks specifically for a report. With a different paradigm tapping into implicit metacognitive processing (e.g. wagering) would we expect similar results? Or do the authors think LPFC involvement is specific for reportable aspects of metacognition? I see that the recent paper by Shea and Frith (2019) is cited, but a paragraph detailing the assumed theoretical relationship seems warranted.

2. Also, there seems to be an inflation of terms used to refer to metacognitive processes: 'metacognitive efficiency', 'metacognitive sensitivity', 'metacognitive awareness', 'metacognitive insight', 'metacognitive ability', and even 'metacognitive awareness sensitivity'... Some are used interchangeably, leading to confusion. Keep what's needed and clarify what refers to as early as possible in the intro. Metacognitive ability/sensitivity, efficiency, and awareness seems what's all needed.

Comments on methods and results:

3. The power analysis revealed 19 as the necessary number of participants, but a total of 34 were recruited. Was a stopping rule employed? Just a question.

4. I wonder if there was a difference of difficulty between tasks. D' results, as I could tell from the graphs, look comparable... How about reaction times?

5. As Type 2 AUC as well as meta- d' measures are susceptible to performance differences, one common method utilized to circumvent any such contamination is to fix the performance levels across participants, e.g. using a staircase paradigm. I am not suggesting authors should collect more data, as I realize how cumbersome this particular procedure is... But perhaps it would at least be enlightening to see performance variability of participants individually and across tasks. I realize meta- d' - d' is the best measure avoiding this very problem, but nevertheless...

6. That said, I wonder why the authors did not opt for a within subjects repeated measures ANOVA including task (orientation/emotion) as an additional factor. This would allow for a direct comparison

of results across tasks.... There is a sentence indicating that analysis is indeed performed, and the effect was observed only with the outcome measure metacognitive efficiency (meta-d'-d') on p. 11.

7. As a final, minor, point, the paragraph on p. 8 referring to the supplementary figure belongs with the figure. So perhaps move the paragraph over to that section or remove that analysis altogether as it doesn't seem to add much to the findings.

Response to Reviewers

We thank the Editor for the opportunity to revise our manuscript “Perceptual metacognition of human faces is causally supported by function of the lateral prefrontal cortex”, and the Reviewers for their thoughtful comments, questions, and suggestions. As a result of addressing them, our revised paper has been strengthened, our findings bolstered by additional analysis, and better contextualized within the existing literature. Below, the Reviewers’ comments are shown in indented *italicized* font, our replies are in regular 12 pt. font, and changes to the manuscript are in **bold**. We are grateful for the opportunity to revise and improve our work, and we look forward to hearing from you.

Regina Lapate, Jason Samaha, Bas Rokers, Brad Postle, & Richard Davidson

Reviewer #1 (Remarks to the Author):

In this study, Lapate et al., investigate the causal effect of TMS to LPFC on metacognitive functioning. Specifically, they find that interfering with LPFC activity impairs metacognitive awareness for the orientation (but not emotion) of human faces. I appreciate their completely within-subjects study design which allows for clean comparisons between different experimental conditions and their efforts to collect a relatively large sample of participants. Their results also potentially suggest a functional dissociation between the neural substrates which support metacognition for low-level stimulus features and those which support metacognition for emotion. I think that this dissociation is very interesting and can further inform the domain specificity vs generality debate on metacognition. Their finding of an interaction between TMS condition and contrast level for the face orientation task is also quite clear and convincing. Overall, their study contributes to existing evidence on the causal involvement of PFC in perceptual metacognition and extends these findings to perception of complex and ecologically relevant stimuli such as human faces.

We thank Reviewer 1 for these supportive comments!

My main concerns are related to the stability of their results, due to low trial counts. I think if the authors can demonstrate the robustness of their findings to alternative methods of analysis, their overall results can be considered more reliable and convincing. I outline my concerns in more detail below:

1. In their methods, they mention that the six presented levels of stimulus contrast consisted of 24 trials each. Given that subjects performed at an accuracy of 75% near their threshold, this would only give them 6 error trials on average, which are critical for computing metacognitive measures such as Type-2 AUC and meta-d’ (which essentially rely on estimates of false alarms). Due to this issue, I am seriously concerned about how reliable their estimates of metacognitive ability are and whether their effects are being biased by applying corrections to the zero cell counts (for example, for incorrect trials with a visibility of 4). In this regard, can they report whether the observation of zero cell counts remained the same or varied between the two TMS conditions?

I can also think of a couple of ways in which the authors can support their findings by demonstrating the reliability of their results. One might be to increase the trial counts for each cell by doing a median split on the confidence scale. In this way, some of the zero-count issues for extreme cells can be avoided. Although using a 4-point scale certainly affords better resolution of confidence, given that trial number can be problematic, I think that in this case, such an analysis might be useful to compound their results. Second, I would suggest pooling contrasts which are close to (or flanking) the threshold for each subject to increase their trial counts. Again, I am aware of the potential disadvantage of pooling contrasts (Rahnev and Fleming (2019) have demonstrated that doing this can inflate estimates of metacognitive ability). However, given that all the subjects experienced the same set of contrasts for all conditions, pooling nearby contrast levels should not confound their results.

We thank Reviewer 1 for raising this important concern and giving us the opportunity to clarify our approach and examine the robustness of our findings. Earlier implementations of meta-*d'* algorithms often handled zero-cell counts with edge correction (Hautus, 1995; Maniscalco & Lau, 2012), and were prone to be biased in those cases. In contrast, the more recently developed Bayesian implementation of meta-*d'* ((Fleming, 2017), <https://github.com/metacoglab/HMeta-d>), which we used in our work, does not rely on zero-count correction (zero counts are handled naturally by a generative model; P. 2, bullet point 3 of Fleming, 2017). This method has been shown to be robust to low trial numbers (as low as $n=20$) in comparison to earlier (e.g. MLE) meta-*d'* implementations (e.g. Figs. 5-8 in Fleming, 2017).

We regret not specifying this in the original Methods section of our paper, which we now do: P. 26: **“To compute meta-*d'*, we used the single-subject Bayesian meta-*d'* algorithm, which does not use cell count correction and thus is robust to low trial numbers relative to earlier meta-*d'* implementations (Fleming 2017).”**

In addition, as shown below, zero counts did not differ between LPFC vs Control cTBS conditions: Orientation discrimination task, across PAS categories: Poisson $B = -0.0198$, $p = 0.86$; per PAS ROC split: [1 vs. 2,3,4] Poisson $B < 0.01$, $p = 0.94$; [1,2 vs. 3,4] Poisson $B = -0.0749$, $p = .61$, and [1,2,3 vs. 4], Poisson $B = -0.0132$, $p = 0.93$. Likewise, for the emotion discrimination task, zero counts did not differ by cTBS condition: across PAS categories, Poisson $B = 0.1335$, $p = 0.56$; per PAS Type 2 ROC split: [1 vs. 2,3,4] Poisson $B = 0.0396$, $p = 0.78$; [1,2 vs. 3,4] Poisson $B < 0.01$, $p = 0.99$; [1,2,3 vs. 4] Poisson $B = 0.0355$, $p = 0.82$.

Orientation task: total # of zero cell counts by cTBS condition (LPFC or S1) & PAS rating (1-4)

	1	2	3	4
d	14	5	16	36
s	11	4	22	37

Emotion task: total # of zero cell counts by cTBS condition (LPFC or S1) & PAS rating (1-4)

	1	2	3	4
d	29	4	20	45
s	24	3	21	41

We now add this to the paper:

P. 26: “Zero counts did not differ between TMS conditions, $ps > .55$ ”

Importantly, results obtained using the non-parametric Type 2 ROC AUC corroborated results obtained from the Bayesian meta- d' model (both at participants' near-threshold contrast, as well as across contrasts). Moreover, the discrepancy between participants' objective performance and subjective visibility (i.e. the source of breakdown of metacognitive awareness following cTBS to LPFC) was also evident in the complementary, non-model based analysis shown in *Supplementary Figure 1*, which also does not rely on correction for zero cell counts.

Following the Reviewer's suggestions, we further examined the robustness of our findings in the following two ways: (a) by performing a median split in the confidence ratings and (b) by pooling across more than one contrast near participants' threshold, as described next.

- (a) A median split on participants' PAS ratings largely replicated our findings. First, at participants' near-threshold contrast, metacognition of face orientation (metacognitive sensitivity and efficiency) was significantly reduced following cTBS to LPFC (vs. Control/S1): Type 2 AUC $t(27) = -3.11, p = 0.004$; meta- $d' t(27) = -2.92, p = 0.006$; meta- $d' - d' t(27) = -2.46, p = 0.02$ (**Figure R1**). As before, inhibitory cTBS to LPFC attenuated orientation metacognition without impacting overall discrimination accuracy or subjective visibility, as evidence by the n.s. impact of cTBS on d' ($p = .8$) and PAS ($p = .23$). In the emotion discrimination task, cTBS continued to not impact metacognition, Type 2 AUC $t(31) = -1.38, p = .17$; meta- $d' t(31) = -0.2, p = .85$; meta- $d' - d' t(31) = -1.97, p = 0.06$. Likewise, emotion discrimination performance (d' ; $p = .16$) and subjective visibility (PAS; $p = .31$) remained unchanged following cTBS to LPFC.

Figure R1: Orientation metacognition by cTBS (PAS Median Split; near-threshold contrast)

We also repeated this analysis across contrasts. AUC estimates from a single ROC point (as is the case with a median split) have the disadvantage of underestimating metacognition values (Bamber, 1975), an effect that we observed in our data: Metacognition estimates were numerically reduced, particularly at the highest contrast levels (see **Figure R2** for a side-by-side comparison of the Type 2 AUC), thereby constraining the range and altering their distribution across contrasts. Perhaps for this reason, the interactive effect of cTBS by contrast on orientation metacognition no longer reached significance using the median split approach, with the exception of meta- $d' - d'$ (**Figure R3**), as follows: Type 2 AUC $p = .4$; meta- $d' p = .36$; meta- $d' - d' W = 23.01, p < 0.01$. Effects in the emotion task were consistent with our original findings,

such that cTBS to LPFC did not change overall face emotion metacognition, Type 2 AUC $p = 0.49$; meta- $d' p = .2$; meta- $d' - d' p = .1$.

Figure R2: Comparison of Type 2 AUC across contrasts (original 4-level PAS vs. median split). AUC estimates are attenuated using a median split (single ROC point) approach.

Figure R3: Orientation metacognition by cTBS (PAS Median Split; across contrasts)

(b) Following Reviewer 1’s suggestion, we also combined data across 2 contrasts closest to participants’ threshold to examine performance near-threshold. Our strategy was to collapse across their closest-to-threshold contrast and the next *highest* contrast¹, as this strategy allowed us to combine data in a consistent manner for all participants in the orientation task, and for all but $n=2$ participants in the emotion discrimination task (those $n=2$ participants’ threshold contrast was the highest one, so for them we pooled data using the next closest (i.e. lower) contrast). This approach, which increased our trial count from $n=24$ to $n=48$, replicated several of our major findings: in the orientation task, cTBS reduced metacognitive sensitivity: Type 2 AUC $t(27) = -2.66, p = 0.013$; meta- $d' t(27) = -2.10, p = 0.045$; although not efficiency meta- $d' - d' t(27) = -1.45, p = 0.15$ (**Figure R4**). As before, inhibitory cTBS to LPFC attenuated orientation metacognitive sensitivity without impacting overall discrimination accuracy or subjective visibility, as evidence by the n.s. impact of cTBS on $d' (p = .72)$ and PAS ($p = .17$).

¹ We pooled over 2 contrasts because $n = 11$ subjects’ threshold contrast in the orientation task was the lowest contrast we used, thus precluding the possibility of including flanking contrasts.

Figure R4: Orientation metacognition by cTBS (collapsed across 2 near-threshold contrasts)

For the emotion discrimination task, collapsing across contrasts unexpectedly resulted in an overall increase in subjective visibility (PAS) following cTBS to LPFC compared to S1 ($t = 4.73$, $p < .001$), while emotion discrimination remained unchanged ($d' = p = .2$); one metric of metacognition was sensitive to this discrepancy between objective and subjective metrics, Type 2 AUC $t(31) = 1.5$, $p = .14$; $meta-d' t(31) = 2.23$, $p = .03$; $meta-d' - d' t(31) = .6$, $p = .5$.

In summary, the impact of cTBS on metacognition of *face orientation* was replicated at participants' near-threshold contrast using the PAS median split approach (Type 2 AUC, $meta-d'$, $meta-d' - d'$) as well as using the pooled contrast approach (for both metrics of metacognitive sensitivity; Type 2 AUC & $meta-d'$). In general, cTBS did not impact metacognition of face emotion, with the exception of the pooled contrast approach for one metric only ($meta-d'$). Collectively, these additional analyses converge to suggest that LPFC function plays a causal role in promoting metacognition of low-level (i.e. orientation) features of complex social stimuli.

2. In Figure 2, they report comparisons of metacognitive measures between the two TMS sites, analyzed only for stimuli at the contrast threshold. The difference between metacognitive efficiency ($meta-d' - d'$) for the two conditions is not significant, which makes me wonder whether some of the changes in $meta-d'$ are being driven by changes in d' . Although they do report d' comparisons separately for the 6 contrast levels, I think it is more important to show these comparisons for the threshold stimuli, because it is for these stimuli that they observe the strongest effect on metacognitive awareness.

We had originally reported d' results at the near-contrast threshold as part of *Footnote 3*, which we have now moved into the main text for clarity. In addition, following the Reviewer's comment, we now include a figure of d' and subjective visibility (PAS) at participants' near-threshold contrast (*Supplementary Figure 2*).

P. 27 : "As mentioned in the *Results*, cTBS did not reliably impact stimulus discrimination performance (d') in either the orientation or emotion task when examined across contrasts ($ps > .4$), nor did it significantly change the contrast which was closest to participants' threshold performance ($ps > .29$)—therefore, we estimated participants' near-threshold face contrast collapsing across cTBS conditions as to increase threshold-estimation

reliability. Note, however, that when examining performance at the near-threshold contrast in the face emotion task, a modest impact of cTBS was observed on d' , $t(31) = 2.12$, $p = .043$, $d = .37$ (accounted for in the metacognitive efficiency estimate $meta-d' - d'$). Near-threshold performance (d') in the face orientation task was not impacted by cTBS, $t(27) = 0.24$, $p = .81$, $d = .04$ (Supplementary Figure 2).”

Supplementary Figure 2. Stimulus discrimination accuracy (d') and subjective visibility (PAS) are plotted as a function of cTBS for face orientation (a & b) and emotion (c & d) discrimination tasks at participants' near-threshold contrast. cTBS to LPFC did not reliably change participants' accuracy (a) or subjective visibility (b) during face orientation discrimination. In the emotion discrimination task, accuracy (c) was slightly larger following cTBS to LPFC, while subjective visibility (d) remained unchanged. Error bars represent within-subjects standard errors (Morey, 2008).

Relatedly, since the LPFC they targeted also covers the dorsolateral prefrontal cortex (BA46), I am curious whether they observed any changes in mean visibility (since previous studies have shown an effect of DLPFC stimulation of mean confidence/visibility as well). If this effect is absent, could the authors comment on why they think this difference may be arising?

We did not observe changes on average visibility following cTBS to LPFC. The area we targeted is located in the mid-DLPFC, but posterior to BA46: in the IFS, between BA44 and BA9/46v (Neubert, Mars, Thomas, Sallet, & Rushworth, 2014; Sallet et al., 2013). The latter prefrontal area is consistent with the cytoarchitectonic division of the estimated left-DLPFC site targeted in Rounis et al (2010) (Rounis, Maniscalco, Rothwell, Passingham, & Lau, 2010). Our data and the data from Rounis et al. 2010 are in agreement in showing that 20-sec cTBS (offline TMS) to mid-DLPFC attenuates metacognitive awareness (i.e. the correspondence between accuracy and subjective visibility/confidence) without necessarily producing a main effect on subjective visibility (see also (Fleming, Ryu, Golfinos, & Blackmon, 2014)).

A recent TMS study did produce a confidence main effect (reduction) by administering online (3-pulse) repetitive TMS to DLPFC immediately after stimulus presentation (Shekhar & Rahnev, 2018). We agree with Shekhar & Rahnev's (2018) suggestion that online TMS might be better suited than offline TMS to alter overall confidence (i.e., less susceptible to adaptation). Moreover, both high and low confidence multivariate signals are mixed within LPFC (Cortese,

Amano, Koizumi, Kawato, & Lau, 2016), which is likely to give rise to inter-study variability in main effects. Finally, we believe that there are important neuroanatomical differences between the DLPFC TMS site targeted by Shekhar & Rahnev 2018 vs. the one targeted by Rounis et al. 2010 & our own, which we describe in detail below, and in our discussion section, in response to Reviewer 1's question #3.

3. Lastly, in the introduction they mention a number of motivating factors for their study such as the controversy surrounding the causal status of PFC in metacognition, the directionality of its functioning and the debate on whether metacognition is domain general or specific. However, I feel that their discussion of their findings does not fully address how these debates or questions are being informed by their current study. For example, they discuss that the controversy surrounding the findings of Rounis et al (2010) may be attributed to their lack of MRI guided neuronavigation. However, they also cite other TMS studies which have used such anatomically/functionally guided neuronavigation techniques for application of TMS (Rahnev et al., 2016; Shekhar and Rahnev, 2018). So, if they do want to address this controversy in their discussion, I think it would be important to include it within the context of all these existing studies and specify how their study can additionally contribute to resolving the debate.

We thank Reviewer 1 for the invitation to better situate our findings in relation to the current literature. We have carefully revised our discussion section with that goal. We believe that our study supports a causal role for mid-DLPFC in metacognition (be it as a precursor to the confidence/ metacognitive computation, or as a site of the confidence computation itself). This contrasts with conclusions from Rahnev et al., 2016 & Shekhar and Rahnev, 2018—in their work, TMS to anterior DLPFC *increased* metacognitive awareness (Rahnev et al., 2016/offline cTBS protocol) or *decreased* confidence without altering metacognitive awareness (Shekhar and Rahnev, 2018/online rTMS protocol) (Rahnev, Nee, Riddle, Larson, & D'Esposito, 2016; Shekhar & Rahnev, 2018). The latter study and associated computational model led to the proposal that DLPFC may serve to simply *relay* the strength of sensory signals to anterior prefrontal cortex (aPFC), where aPFC is posited to transform the information received from DLPFC into a confidence judgment, subserving metacognition (Shekhar & Rahnev, 2018).

While our conclusion – that mid-DLPFC function *does* play a causal role in metacognitive awareness – contrasts with theirs, our findings are not necessarily inconsistent with their proposal: altering DLPFC function may result in a ‘corrupted’ sensory signal being conveyed to aPFC, thereby indirectly impacting the confidence computation, and metacognitive awareness. Nonetheless, it is important to consider anatomical and functional connectivity fingerprints of DLPFC sites targeted across these different studies, and how they may differentially impact metacognition.

The DLPFC coordinate we targeted (originally chosen based on our prior work examining visual awareness changes during emotional processing (Lapate et al., 2016)) was recently identified as a reliable correlate of metacognition in the largest neuroimaging meta-analysis on this topic to

Figure R5.

date (Vaccaro & Fleming, 2018) (**Figure R5**). This left IFS site is located between 9/46v and BA44 (Neubert et al., 2014; Sallet et al., 2013). Rounis et al. (2010)'s left-LPFC location, while dorsal, was similarly posterior, in 9/46v (Sallet et al., 2013). Contrastingly, the DLPFC

targeted by Shekhar & Rahnev 2018 is anterior (~.6cm) and medial (>1.3cm) relative to those, putatively located in a different cytoarchitectonic area (BA8B) (Sallet et al., 2013).

This anatomical discrepancy appears meaningful at the functional network level. A resting state functional connectivity meta-analysis (n=1000 subjects; (Yeo et al., 2011)) conducted using those DLPFC coordinates reveals that ours and Rounis's et al. (2010)'s DLPFC targets were characterized by (1) overlapping functional connectivity profiles in lateral frontal regions of the frontoparietal network and (2) reliable connectivity with visual (lateral occipital) cortex (**Figure R6**). This contrasts with Rahnev's DLPFC targets, which (1) coupled with the frontoparietal network, but in largely non-overlapping frontal sites relative to our and Rounis et al's (2010) study; and (2) do not show reliable visual cortical coupling at this (low) threshold; instead, they coupled with the posterior cingulate cortex node from the default mode network (known to track internal states; (Raichle, 2015)) (**Figure R6**). While speculative, this network affiliation discrepancy raises the possibility that increased visual cortical-DLPFC coupling may underlie the differential contribution of BA 9/46v (mid-DLPFC) to perceptual metacognition (our study and Rounis et al. 2010) vs. BA8B (anterior-medial DLPFC) to confidence ratings (Shekhar & Rahnev, 2018).

Figure R6.

The above comparison of functional connectivity profiles of previously targeted DLPFC sites underscores not only the importance of anatomical precision, but also the little-understood and stark heterogeneity of subnetworks found *within* DLPFC. Moving forward, we believe that dissecting the functional specialization of these separable prefrontal networks will prove critical for a fuller understanding of the neural architecture of metacognition.

We have included Figure R.6 as *Supplementary Figure 3* in the revised manuscript, and we summarized the points above in the discussion, as follows:

P.14: “(...) However, Rahnev et al. (2016) targeted a DLPFC region anterior relative to the one targeted in the current study and in Rounis et al. (2010), which recapitulated metacognitive-enhancing effects that same group has found when targeting anterior PFC (aPFC/BA10) (Rahnev et al., 2016; Shekhar & Rahnev, 2018).”

P. 15: “Of note, two prior studies using MRI-guided TMS (Rahnev et al., 2016; Shekhar & Rahnev, 2018) concluded that the mid-LPFC (DLPFC) may not contribute directly to perceptual metacognition. In Shekhar & Rahnev (2018), online TMS to DLPFC lowered confidence ratings, without changing metacognitive awareness, whereas both offline and online TMS to aPFC have been found to increase metacognitive awareness (putatively by reducing metacognitive noise) (Rahnev et al., 2016; Shekhar & Rahnev, 2018). In light of those data and a recent computational model, DLPFC was proposed to simply relay the strength of sensory evidence to aPFC (BA10), which would in turn transform the information received from DLPFC into a confidence judgment, and be the proximal site subserving metacognition (Shekhar & Rahnev, 2018). Our findings are not incompatible with their proposal: if mid-DLPFC-originated sensory evidence precede confidence computations in aPFC, local perturbations may ultimately cascade in altered metacognitive estimates computed by a later node. Nonetheless, it is important to note the neuroanatomical heterogeneity across DLPFC sites targeted in these studies. The left-DLPFC regions targeted here and in Rounis et al. (2010) are posterior and lateral to right DLPFC sites used in Shekhar & Rahnev (2018) and Rahnev et al. (2016). Accordingly, their functional connectivity fingerprints are distinct (*Supplementary Figure 3*), with our site and Rounis et al.’s (2010) showing greater DLPFC-visual cortical coupling and a notably divergent profile of frontal network affiliation compared to Shekhar & Rahnev (2018) and Rahnev et al. (2016); whether these distinct network affiliations account for the distinct functional contributions revealed by causal perturbation of those DLPFC sites remains to be determined. Moving forward, dissecting the functional specialization of these separable prefrontal networks will likely prove critical for a thorough understanding of the neural architecture of metacognition.”

We have also expanded our discussion paragraph addressing domain specificity:

P. 16: “*Is mid-LPFC function in metacognition domain-specific?*”

In this study, we probed metacognition of faces using an approach that is well aligned with the extant literature on perceptual (visual) metacognition, which has often adopted stimulus types and discrimination tasks in which orientation was a core discerning feature (Rounis et al., 2010; Shekhar & Rahnev, 2018). In order to glean insight into the domain generality of lateral prefrontal contributions to metacognition of complex social stimuli, we also separately examined metacognition of face *emotion*, a core feature of human faces (Lee, Ruby, Giles, & Lau, 2018; Rouault, McWilliams, Allen, & Fleming, 2018). Our data showed preliminary evidence for a possible dissociation between metacognitive judgments for orientation vs. emotion features of complex face stimuli. In contrast with robust attenuation of metacognition of face spatial orientation following cTBS to LPFC, metacognitive awareness of emotional expressions was largely unaffected by cTBS, suggesting a possible dissociation of the neural substrates supporting metacognition of these two important face features. However, as the test of interaction

between task and cTBS site only reached significance for a metric of metacognitive efficiency ($\text{meta-}d'-d'$) and not for metacognitive sensitivity alone (AUC and $\text{meta-}d'$), we are cautious in interpreting this effect, and hope that it paves the way for future studies. For instance, it is possible that metacognition of emotional expressions relies on re-representations of emotional valence in superior temporal and medial frontal (including interoceptive) circuitry, rather than on occipital-LPFC projections (Reuter et al., 2009). Consistent with this idea, a recent study found that metacognitive awareness of emotional expressions correlated with function and white matter microstructure of the cingulate cortex, and not of LPFC (Bègue et al., 2019). As a rigorous neuroscience of emotional consciousness is in its nascent stages (Brown, Lau, & LeDoux, 2019; Ledoux & Brown, 2017), carefully delineating first and higher-order correlates of human affective encoding and experiences—and testing their causal contribution to conscious emotional states—will be critical avenues for future work.”

Supplementary Figure 3. Neurosynth derived resting-state functional connectivity maps (Yeo et al. 2011) ($N = 1,000$ subjects; MNI space, 2mm isotropic) are shown for DLPFC sites targeted in previous TMS studies of metacognitive awareness, thresholded at $r > .2$. While overlap of functional connectivity fingerprints is observed across studies in parietal cortex, distinct visual-cortical and intra-prefrontal network profiles are noted when comparing Rounis et al. (2010) and the present study vs. Rahnev et al. (2016) and Shekhar & Rahnev (2018).

The functional connectivity maps above can be assessed here:

Lapate et al (current study): https://neurosynth.org/locations/-48_24_20_6/

Rounis et al 2010 (left coordinate): https://neurosynth.org/locations/-40_18_52_6/

Shekhar & Rahnev 2018: https://neurosynth.org/locations/28_30_38_6/

Rahnev et al. 2016 (average coordinate): https://neurosynth.org/locations/38_34_28_6/

Additionally, it may be helpful to include subsections in the discussion for each of these three motivating questions and specifically address how their findings can contribute to our understanding in these areas.

We thank the Reviewer for that suggestion and have added the following subsections in the discussion section:

P. 14-18:

The causal status of mid-LPFC in metacognitive awareness: relation to prior work

The computational function of LPFC in metacognitive awareness

Is LPFC function in metacognition domain-specific?

Limitations and Future Directions

Conclusion

Also, at the very end of the introduction, where the authors report their findings, I think it would also be useful to briefly mention the implications of their findings within the context of their motivation.

We agree, and have added the following note to the end of the intro:

P. 6: “Collectively, these findings contribute to a prior controversial literature by demonstrating that (1) mid-LPFC function plays a causal role in metacognition, a role which (2) extends beyond simple visual stimuli to include introspective reports of naturalistic social stimuli (i.e. faces), while raising the possibility that (3) metacognitive dissociations may occur during the processing of complex social stimuli.”

Minor comments:

1. I think the domain generality vs domain specificity effect in the introduction (p4, line 7) can be expanded a little more for context and clarity, if I am correctly assuming that this is also one of their main research objectives.

Examining the domain specificity vs. generality of mid-LPFC function was a secondary, and relatively exploratory aim of the current experiment; our strongest a-priori hypothesis pertained to orientation judgments given that spatial orientation had been frequently probed in prior work. We are reluctant to expand the section much beyond its current length given that the interaction of task and cTBS site in this experiment was only significant when examining meta- $d' - d'$ (across contrasts), $W = 5.183$ $p = 0.034$ (P. 13). Nonetheless, given the suggestive pattern of results and the paucity of existing data on this, we believe that this is an exciting direction for future work. Thus, we have edited the introduction to briefly raise the issue of domain (and more specifically, feature) generality in the neural correlates of perceptual decision making, as follows:

P. 4: “(...) This approach, coupled with precise TMS neuronavigation, may not only help adjudicate between prior disparate findings, but also help clarify domain specificity in the neural architecture supporting metacognition. For instance, while metacognition of low-level visual features often correlates with function of LPFC ((Rouault et al., 2018), see (Vaccaro & Fleming, 2018) for a meta-analysis), it remains unclear whether metacognition of emotional features relies on the same lateral prefrontal network (Ledoux & Brown, 2017) or instead may rely on a separate medial prefrontal, interoceptive-representing circuitry (Adolphs, 2013; Bègue et al., 2019).”

2. At the beginning of the results section, a brief summary of the methods (task and comparisons) and objective of the study will be useful to anchor the readers.

We thank the Reviewer for this suggestion. We have added the following overview to the beginning of the results section:

P. 8: “**In the following, we examined whether inhibitory cTBS to LPFC (vs Control/S1) modulated metacognition. We examined both metacognitive sensitivity (Type 2 AUC & meta- d') and efficiency (meta- $d' - d'$). Following prior work (Bor, Schwartzman, Barrett, & Seth, 2017; Rounis et al., 2010), we examined metacognition at the contrast closest to participants' detection threshold (75%) using paired-samples t-tests. Next, using the full data obtained with the method of constant stimuli, we also probed whether inhibitory cTBS to LPFC altered metacognition independently of stimulus strength (i.e. contrast) using a repeated-measures analysis with cTBS (2) and stimulus contrast (6) as within-subjects factors. We first describe the results pertaining to the face orientation task given its direct relevance to prior work (Bor et al., 2017; Rahnev et al., 2016; Rounis et al., 2010; Shekhar & Rahnev, 2018), followed by results pertaining to the face emotion identification task, and a formal comparison between the two tasks.**”

3. In the reporting of results on p10, why are the actual p-values not being reported for the analyses on the near threshold stimulus contrasts? I think there should only be a single p-value for paired t-tests in these analyses.

The t -tests reported on page 10 (at the near-threshold contrast for the emotion discrimination task; Type 2 AUC, meta- d' and meta- $d' - d'$) are now each followed by their p -values:

P. 11: “(...) **inhibitory cTBS to LPFC did not impair metacognition of face emotion. This null finding was consistent across all metrics of metacognitive awareness, whether examined at near-threshold stimulus contrasts, Type 2 AUC $t(31) = .7, p = .49, d = .12$; meta- $d' t(31) = .626, p = .54, d = 0.11$; meta- $d' - d' t(31) = -1.15, p = 0.26, d = .2$ (Fig. 4 a-c), or whether examined across all contrasts, Type 2 AUC $W = 1.277, p = .33, \eta_p^2 = .191$; meta- $d' W = 4.728, p = .547, \eta_p^2 = .136$; meta- $d' - d' W = 7.61, p = .286, \eta_p^2 = .202$ (Fig. 4 d-f).**”

4. Their results suggest an interpretation that PFC is involved in perception of low-level features of faces but not their emotional content. However, I am not very convinced by the emphasis on ‘orientation of emotional faces’ (abstract, line 11) because their study design does not support comparisons of emotional faces against neutral faces. More

specifically, without knowing how their results might have differed for orientation discrimination on neutral faces, the emphasis on emotional faces does not seem very meaningful here.

We thank the Reviewer for this important comment; we agree, and have removed “emotional” from that sentence in the abstract (and from everywhere else face orientation results had been described in a similar manner).

5. Can the authors add some details explaining how they decided on the 6 different contrast levels used for the experiment?

Yes, we have added the following note to the *Methods* section:

P. 24: “The contrasts adopted in our study were chosen empirically, initially based on the behavioral performance of three of the authors (RCL, BR, JS), followed by iterative refinement based on behavioral piloting using a total of n=13 naïve observers (undergraduate students at the University of Wisconsin-Madison). Our goal was to find a set of contrasts that reliably captured the psychometric curve of most or all observers, including performance near threshold (75%). The set of contrasts used here was tested on 3 of the authors (RCL, BR, JS) and n=9 naïve observers, and captured threshold performance of n=11/12 of them.”

6. Metacognitive efficiency, measured by meta- d' - d' , is thought control for the effects of variation in task performance. So, I wonder why there is so much variation in meta- d' - d' across the 6 different contrast conditions. Can the authors comment on this?

We thank the Reviewer for this interesting question—as noted by the Reviewer and seen in Figures 2f & 4f, meta- $d' - d'$ plotted across stimulus contrasts reveals a u-shaped curve: metacognitive efficiency estimates at middle contrasts are relatively similar to one another, whereas the lowest and 2 highest contrasts are characterized by higher metacognitive efficiency. Based on prior theoretical models and existing data, we believe that there are two independent sources of variation accounting for this pattern: (a) initial downswing (lowest contrast): Recent work by Bang, Shekhar & Rahnev (2019) demonstrates that, perhaps counterintuitively, a hierarchical model of metacognition predicts that higher sensory noise can result in larger estimates of metacognitive efficiency (Bang, Shekhar, & Rahnev, 2019). In brief, their hierarchical model assumes two sources of noise: sensory and metacognitive. Metacognition (e.g. meta- d') can be corrupted by *both* types of noise, whereas perception (d') is corrupted only by sensory noise. As sensory noise levels change across contrasts (e.g. stimulus becomes clearer) this dissociation gives rise to non-constancy in metacognitive efficiency (meta- $d' - d'$) values. Higher sensory noise (e.g. low contrast) will greatly impact d' , but will not impact meta- d' to the same degree; thereby resulting in a larger meta- $d' - d'$ at lower contrasts. (b) upswing (highest contrasts): Theoretically, meta- $d' - d'$ in the ideal observer should be as high as 1 (i.e., perfect metacognition)—but in practice, “hyper metacognitive sensitivity” (meta- $d' - d' > 1$) has often been observed empirically ((Fleming & Daw, 2017) *Figure 8E* for a compilation of recent data). What happens is, as contrast increases, errors tend to be driven by motoric or attentional lapses, as opposed to sensory lapses. In essence, at high contrasts errors

are less frequent, but more obvious: the introspective observer realizes potential motoric/attentional lapses when the sensory information is clear. A Bayesian model that incorporates post-decisional factors accounts for this phenomenon: Reliable error detection gives rise to hyper metacognitive sensitivity (i.e. $\text{meta-}d' - d' > 1$) when contrast increases (Fleming & Daw, 2017).

Reviewer #2 (Remarks to the Author):

Manuscript#: COMMSBIO-19-1456-T

Summary: The study aims to investigate a possible causal link between LPFC activity and metacognitive ability, as suggested previous work in the literature. By administering MRI navigated cTBS protocol with a paradigm that varies the noise in the visual stimuli, the study demonstrates that LPFC supports metacognitive ability in perceptual metacognition of face orientation, but not of emotional expression of faces.

Overall Impression: I much enjoyed reading this manuscript. It's a very well written paper on a study that employed very clean manipulations bringing together different methodologies to address a very specific question. It's a novel and valuable contribution to the literature. My own work will certainly benefit from it.

We thank Reviewer 2 for these supportive comments!

Comments on Introduction and discussion:

1. Authors would want to incorporate a discussion in the intro (and perhaps in the discussion as well) to explain the link between consciousness and metacognition, and what's exactly referred to with 'metacognitive awareness'. Metacognition can also operate implicitly, and the prompt used in this paradigm ('how clear was your experience') asks specifically for a report. With a different paradigm tapping into implicit metacognitive processing (e.g. wagering) would we expect similar results? Or do the authors think LPFC involvement is specific for reportable aspects of metacognition? I see that the recent paper by Shea and Frith (2019) is cited, but a paragraph detailing the assumed theoretical relationship seems warranted.

We thank the Reviewer for inviting us to explore this important issue. We have expanded the following paragraph in the introduction clarifying our definition and position on this:

P. 2 "Prominent theories of consciousness, such as the Global Workspace Theory and Higher Order Theories (Dehaene, Lau, & Kouider, 2017; Shea & Frith, 2019), posit a relationship between metacognition and conscious perception. For instance, metacognitive awareness, the ability to accurately monitor one's internal sensory experiences, has been proposed to be a precursor to consciousness (Brown et al., 2019). Relatedly, some argue that all conscious percepts may be inherently imbued with a metacognitive component, which would facilitate their integration in the global workspace and permit optimal decision making across domains (Shea & Frith, 2019) (but see also (Dehaene et al., 2017))."

While some of these theories disagree on the precise relationship between metacognition and consciousness, they converge in proposing that function of the prefrontal cortex (PFC) plays a critical role in promoting metacognitive awareness and conscious perception, regardless of whether the content of conscious awareness has to be explicitly reported (Kapoor, Besserve, Logothetis, & Panagiotaropoulos, 2018; Panagiotaropoulos, Deco, Kapoor, & Logothetis, 2012) (but see also (Tsuchiya, Wilke, Frässle, & Lamme, 2015)).”

2. Also, there seems to be an inflation of terms used to refer to metacognitive processes: ‘metacognitive efficiency’, ‘metacognitive sensitivity’, ‘metacognitive awareness’, ‘metacognitive insight’, ‘metacognitive ability’, and even ‘metacognitive awareness sensitivity’ ... Some are used interchangeably, leading to confusion. Keep what’s needed and clarify what refers to as early as possible in the intro. Metacognitive ability/sensitivity, efficiency, and awareness seems what’s all needed.

We agree with Reviewer 2 that this term inflation is unhelpful. We have retained the terms *metacognitive awareness/ability* (to refer to the overall conceptual construct), *metacognitive sensitivity* (to refer to Type 2 AUC & meta- d' measures) and *metacognitive efficiency* (to refer to the meta- $d' - d'$ metric).

Comments on methods and results:

3. The power analysis revealed 19 as the necessary number of participants, but a total of 34 were recruited. Was a stopping rule employed? Just a question.

We anticipated larger participant attrition and incomplete data than we had—due to the demand of multiple sessions, such as clinical screening, MRI, EEG/TMS session, and the possibility that certain participants would perform consistently below or above chance in the tasks using the method of constant stimuli. Thus, with the goal of retaining a minimum of $n = 19$ participants with useable data across the multiple TMS sessions and psychophysical assessments, we recruited the largest number of participants that we were able to in the year this study was run (prior to the graduation of the first author).

4. I wonder if there was a difference of difficulty between tasks. D' results, as I could tell from the graphs, look comparable... How about reaction times?

The emotion-discrimination task was on average (across contrasts) more difficult than the orientation-discrimination task—i.e., d' across contrasts on Figure 3A is shifted slightly upwards for the orientation relative to the emotion task, which is statistically significant, $W(1,5) = 131.79$, $p < 0.01$. In practice, this issue is circumvented in the analysis that focuses on the near-threshold contrast (as the near-contrast was identified per participant based on 2AFC performance separately for each task; Figures 2a-c and 4a-c, and where there was no difference between d' between the tasks, $p = .1$). Nonetheless, the performance difference between tasks is a limitation when directly comparing metacognitive sensitivity (which does not control for task performance) between the tasks across all contrasts. We now note this limitation in the results section of our paper, where we have included the heading:

P. 12:

“Comparing metacognition across the orientation and emotion tasks:

(...) Even though performance between the two tasks did not differ at participant’ contrast threshold, $p = .1$, it did when the data were examined across all contrasts (Fig. 3a & 3c), $W(1,5) = 131.79, p < 0.001$), thereby limiting the interpretability of metacognitive indices that do not control for task performance when examining the data across contrasts.”

Reaction times did not differ between the two tasks (across contrasts or at the threshold contrast), $p_s > .14$.

5. As Type 2 AUC as well as meta- d' measures are susceptible to performance differences, one common method utilized to circumvent any such contamination is to fix the performance levels across participants, e.g. using a staircase paradigm. I am not suggesting authors should collect more data, as I realize how cumbersome this particular procedure is... But perhaps it would at least be enlightening to see performance variability of participants individually and across tasks. I realize meta- d' - d' is the best measure avoiding this very problem, but nevertheless...

Experimental design provides an inherent tension between making sure all participants see the same stimuli, and equating task difficulty. A staircase procedure is often used; here, we used the method of constant stimuli to be able to examine whether the putative role for LPFC in promoting metacognitive visual awareness would be specific to near-threshold stimuli presentations, or whether it would extend to sub- and supra-threshold stimuli. We circumvent possible performance differences in the following two ways: (1) examining the impact of cTBS on metacognition at the contrast nearest to each participants’ threshold (determined separately per task – which is conceptually similar to finding a contrast threshold with a staircase), and (2) examining *meta- d' - d'* . In our data, the within-subject variability (i.e SD of the within-subject task difference) was smaller ($SD_w=0.292$) than the between-subject variability (i.e. the pooled standard deviation across the two tasks, $SD=0.307$). To help visualize between-subject performance variability per task, below we plot participants’ performance at the near-contrast threshold as a function of task (orientation vs. emotion, colored by cTBS condition):

Figure R7. Task discrimination performance (d') at participants' near-contrast threshold plotted for orientation and emotion discrimination tasks (colored by cTBS condition).

6. That said, I wonder why the authors did not opt for a within subjects repeated measures ANOVA including task (orientation/emotion) as an additional factor. This would allow for a direct comparison of results across tasks.... There is a sentence indicating that analysis is indeed performed, and the effect was observed only with the outcome measure metacognitive efficiency (meta- d' - d') on p. 11.

Prior work on the causal role of LPFC to metacognition has frequently probed metacognition of perceptual decisions involving stimulus orientation (Bor et al., 2017; Rahnev et al., 2016; Rounis et al., 2010; Shekhar & Rahnev, 2018). Therefore, our strongest a-priori hypothesis pertained to the orientation task. Examining metacognition of face emotion was a secondary, and more exploratory, goal of the current study. Thus, we initially present the data separately by task since the results of the orientation task can be more readily integrated with the prior literature, followed by the analysis of face emotion metacognition and a formal comparison between the two tasks.

Following the Reviewer's comment, we have now added a heading for the task comparison step at the end of our results section ("**Comparing metacognition across orientation and emotion domains**") and, following this Reviewer's comment #3, we have supplemented it with stimulus identification performance results compared directly across the two tasks. In addition, following Reviewer 1's suggestion, we include an overview of the order that results are reported in at the beginning of the Results section as to better orient the reader:

P. 8

"(...) We first describe the results pertaining to the face orientation task given direct relevance to prior studies, followed by results pertaining to the face emotion identification task, and a formal comparison between the two tasks."

7. As a final, minor, point, the paragraph on p. 8 referring to the supplementary figure

belongs with the figure. So perhaps move the paragraph over to that section or remove that analysis altogether as it doesn't seem to add much to the findings.

We have moved the paragraph to the Supplementary Results section, and we now summarize the take-home point from that analysis at the end of the preceding paragraph:

P. 9: “(...) The analysis of the association between visibility ratings and stimulus discrimination performance at participants’ near-threshold contrast indicated that subjective visibility following incorrect trials was rated higher following inhibitory cTBS to LPFC, thereby clarifying the nature of the reduced metacognitive awareness observed when LPFC function was altered. (*Supplementary Fig. 1*).”

References

- Adolphs, R. (2013). The biology of fear. *Current Biology*, 23(2), R79-93.
<https://doi.org/10.1016/j.cub.2012.11.055>
- Bamber, D. (1975). The Area above the Ordinal Dominance Graph and the Area below the ROC Graph. *Journal of Mathematical Psychology*.
- Bang, J. W., Shekhar, M., & Rahnev, D. (2019). Sensory noise increases metacognitive efficiency. *Journal of Experimental Psychology: General*. <https://doi.org/10.1037/xge0000511>
- Bègue, I., Vaessen, M., Hofmeister, J., Pereira, M., Schwartz, S., & Vuilleumier, P. (2019). Confidence of emotion expression recognition recruits brain regions outside the face perception network. *Social Cognitive and Affective Neuroscience*, 14(1), 81–95. <https://doi.org/10.1093/scan/nsy102>
- Bor, D., Schwartzman, D. J., Barrett, A. B., & Seth, A. K. (2017). Theta-burst transcranial magnetic stimulation to the prefrontal or parietal cortex does not impair metacognitive visual awareness. *PLoS ONE*, 12(2), 1–20. <https://doi.org/10.1371/journal.pone.0171793>
- Brown, R., Lau, H., & LeDoux, J. E. (2019). Understanding the Higher-Order Approach to Consciousness. *Trends in Cognitive Sciences*, 23(9), 754–768.
<https://doi.org/10.1016/j.tics.2019.06.009>
- Cortese, A., Amano, K., Koizumi, A., Kawato, M., & Lau, H. (2016). Multivoxel neurofeedback selectively modulates confidence without changing perceptual performance. *Nature Communications*, 7, 1–18. <https://doi.org/10.1038/ncomms13669>
- Dehaene, S., Lau, H., & Kouider, S. (2017). What is consciousness, and could machines have it? *Science*, 358(27), 1–7.
- Fleming, S. M. (2017). HMeta-d: hierarchical Bayesian estimation of metacognitive efficiency from confidence ratings. *Neuroscience of Consciousness*, 2017(1), 1–14.
<https://doi.org/10.1093/nc/nix007>
- Fleming, S. M., & Daw, N. D. (2017). Self-evaluation of decision-making: A general bayesian framework for metacognitive computation. *Psychological Review*, 124(1), 91–114.
<https://doi.org/10.1037/rev0000045>
- Fleming, S. M., Ryu, J., Golfinos, J. G., & Blackmon, K. E. (2014). Domain-specific impairment in metacognitive accuracy following anterior prefrontal lesions. *Brain*, 137(10), 2811–2822.
<https://doi.org/10.1093/brain/awu221>
- Hautus, M. J. (1995). Corrections for extreme proportions and their biasing effects on estimated values of d' . *Behavior Research Methods, Instruments, & Computers*, 27(1), 46–51.
<https://doi.org/10.3758/BF03203619>
- Kapoor, V., Besserve, M., Logothetis, N. K., & Panagiotaropoulos, T. I. (2018). Parallel and functionally segregated processing of task phase and conscious content in the prefrontal cortex. *Communications Biology*, 1(1). <https://doi.org/10.1038/s42003-018-0225-1>
- Lapate, R. C., Rokers, B., Tromp, D. P. M., Orfali, N. S., Oler, J. A., Doran, S. T., ... Davidson, R. J. (2016). Awareness of Emotional Stimuli Determines the Behavioral Consequences of Amygdala Activation and Amygdala-Prefrontal Connectivity. *Scientific Reports*, 6, 25826.
<https://doi.org/10.1038/srep25826>
- Ledoux, J. E., & Brown, R. (2017). A higher-order theory of emotional consciousness. *Proceedings of the National Academy of Sciences of the United States of America*, 114(10), E2016–E2025.
<https://doi.org/10.1073/pnas.1619316114>
- Lee, A. L. F., Ruby, E., Giles, N., & Lau, H. (2018). Cross-domain association in metacognitive efficiency depends on first-order task types. *Frontiers in Psychology*, 9(DEC), 1–10.
<https://doi.org/10.3389/fpsyg.2018.02464>
- Maniscalco, B., & Lau, H. (2012). A signal detection theoretic approach for estimating metacognitive sensitivity from confidence ratings. *Consciousness and Cognition*, 21(1), 422–430.
<https://doi.org/10.1016/j.concog.2011.09.021>
- Morey, R. (2008). Confidence intervals from normalized data: A correction to Cousineau (2005). *Tutorials in Quantitative Methods for Psychology*, 4(2), 61–64. Retrieved from

- <http://pcl.missouri.edu/sites/default/files/morey.2008.pdf>
- Neubert, F. X., Mars, R. B., Thomas, A. G., Sallet, J., & Rushworth, M. F. S. (2014). Comparison of Human Ventral Frontal Cortex Areas for Cognitive Control and Language with Areas in Monkey Frontal Cortex. *Neuron*, *81*(3), 700–713. <https://doi.org/10.1016/j.neuron.2013.11.012>
- Panagiotaropoulos, T. I., Deco, G., Kapoor, V., & Logothetis, N. K. (2012). Neuronal Discharges and Gamma Oscillations Explicitly Reflect Visual Consciousness in the Lateral Prefrontal Cortex. *Neuron*, *74*(5), 924–935. <https://doi.org/10.1016/j.neuron.2012.04.013>
- Rahnev, D., Nee, D. E., Riddle, J., Larson, A. S., & D’Esposito, M. (2016). Causal evidence for frontal cortex organization for perceptual decision making. *Proceedings of the National Academy of Sciences*, *113*(21), 6059–6064. <https://doi.org/10.1073/pnas.1522551113>
- Raichle, M. E. (2015). The Brain’s Default Mode Network. *Annual Review of Neuroscience*, *38*(1), 433–447. <https://doi.org/10.1146/annurev-neuro-071013-014030>
- Reuter, F., Del Cul, A., Malikova, I., Naccache, L., Confort-Gouny, S., Cohen, L., ... Audoin, B. (2009). White matter damage impairs access to consciousness in multiple sclerosis. *NeuroImage*, *44*(2), 590–599. <https://doi.org/10.1016/j.neuroimage.2008.08.024>
- Rouault, M., McWilliams, A., Allen, M. G., & Fleming, S. M. (2018). Human Metacognition Across Domains: Insights from Individual Differences and Neuroimaging. *Personality Neuroscience*, *1*, 1–13. <https://doi.org/10.1017/pen.2018.16>
- Rounis, E., Maniscalco, B., Rothwell, J. C., Passingham, R. E., & Lau, H. (2010). Theta-burst transcranial magnetic stimulation to the prefrontal cortex impairs metacognitive visual awareness. *Cognitive Neuroscience*, *1*(3), 165–175. <https://doi.org/10.1080/17588921003632529>
- Sallet, J., Mars, R. B., Noonan, M. P., Neubert, F.-X., Jbabdi, S., O’Reilly, J. X., ... Rushworth, M. F. (2013). The Organization of Dorsal Frontal Cortex in Humans and Macaques. *Journal of Neuroscience*, *33*(30), 12255–12274. <https://doi.org/10.1523/jneurosci.5108-12.2013>
- Shea, N., & Frith, C. D. (2019). The Global Workspace Needs Metacognition. *Trends in Cognitive Sciences*, *23*(7), 560–571. <https://doi.org/10.1016/j.tics.2019.04.007>
- Shekhar, M., & Rahnev, D. (2018). Distinguishing the Roles of Dorsolateral and Anterior PFC in Visual Metacognition. *The Journal of Neuroscience*, *38*(22), 5078–5087. <https://doi.org/10.1523/jneurosci.3484-17.2018>
- Thomas Yeo, B. T., Krienen, F. M., Sepulcre, J., Sabuncu, M. R., Lashkari, D., Hollinshead, M., ... Buckner, R. L. (2011). The organization of the human cerebral cortex estimated by intrinsic functional connectivity. *Journal of Neurophysiology*, *106*(3), 1125–1165. <https://doi.org/10.1152/jn.00338.2011>
- Tsuchiya, N., Wilke, M., Frässle, S., & Lamme, V. A. F. (2015). No-Report Paradigms: Extracting the True Neural Correlates of Consciousness. *Trends in Cognitive Sciences*, *19*(12), 757–770. <https://doi.org/10.1016/j.tics.2015.10.002>
- Vaccaro, A. G., & Fleming, S. M. (2018). Thinking about thinking: A coordinate-based meta-analysis of neuroimaging studies of metacognitive judgements. *Brain and Neuroscience Advances*, *2*, 239821281881059. <https://doi.org/10.1177/2398212818810591>

Reviewers' comments:

Reviewer #1 (Remarks to the Author):

I thank the authors for thoroughly addressing all the comments and for their thoughtful discussions. I particularly appreciate their analysis of functional connectivity for the TMS sites targeted by different studies, which opens up interesting possibilities to test in the future. I think the additional analyses have strengthened their main conclusions. I would like to make a few minor comments below:

1. I think the authors should report and analyze the zero counts separately for correct and incorrect trials because it is differences in high confidence for incorrect trials between conditions (due to lower counts) that can lead to biases. Additionally, the Poisson test seems a bit cumbersome as it requires collapsing the scale into all possible binary scales. I was wondering whether a chi squared (goodness of fit) test would be more appropriate to check for differences in the distribution of confidence among all eight possible confidence responses (i.e. confidence of 1-4 separately for correct and incorrect responses). Also, I think the name of the tests performed for these analyses should be mentioned in the manuscript as well.

2. The term 'metacognitive dissociations' (p6) does not seem to be a commonly used term and I was initially confused as I thought they were referring to dissociation between confidence and stimulus perception, which was not the case. Can the authors expand or rephrase this conclusion in more clear terms?

3. In Bang et al (2019), the changes in sensory noise were induced either by learning or by the mixture of trials from different levels of contrasts (which increased the between-trial sensory noise). It is not clear why sensory noise should vary between different contrasts when we use trials of the same contrast level. I wonder whether these results can be further explained by learning. If participants were able to learn during the course of the task for middle (near-threshold) contrasts, it may have lowered their sensory noise and their metacognitive efficiency. However, for contrasts much below their threshold, if no learning was possible, it could have resulted in higher sensory noise and metacognitive efficiency. I recognize that all these arguments are speculative. Nevertheless, I think that the authors' entire discussion about the variability of metacognitive efficiency across contrasts would serve as a useful reference for future studies and should also be included in their discussion.

Response to Reviewers

We thank the Editor for the opportunity to revise our manuscript “Perceptual metacognition of human faces is causally supported by function of the lateral prefrontal cortex”, and Reviewer I for their thoughtful additional comments. We believed our manuscript has been further strengthened by incorporating the Reviewer’s feedback in this second round of revision. Below, the Reviewer’s comments are shown indented, our replies are in **regular 12 pt. blue** font, and changes to the manuscript are in **bold**. We are grateful for the opportunity to further revise and improve our work, and we look forward to hearing from you.

Regina Lapate, Jason Samaha, Bas Rokers, Brad Postle, & Richard Davidson

Reviewer #1 (Remarks to the Author):

I thank the authors for thoroughly addressing all the comments and for their thoughtful discussions. I particularly appreciate their analysis of functional connectivity for the TMS sites targeted by different studies, which opens up interesting possibilities to test in the future. I think the additional analyses have strengthened their main conclusions.

We thank Reviewer 1 for these supportive comments!

I would like to make a few minor comments below:

1. I think the authors should report and analyze the zero counts separately for correct and incorrect trials because it is differences in high confidence for incorrect trials between conditions (due to lower counts) that can lead to biases.

We thank Reviewer 1 for this suggestion; we now report zero counts separately for correct and incorrect trials at participants’ near-threshold contrast as a function of cTBS condition (LPFC vs. S1).

Table S1: Orientation task: total # of zero cell counts by cTBS condition (LPFC or S1), stimulus-discrimination accuracy (Correct or Incorrect), & PAS rating (1-4)

	Correct Trials				Incorrect Trials			
	1	2	3	4	1	2	3	4
LPFC	5	1	5	14	9	4	11	22
S1	6	0	3	12	5	4	19	25

Table S2: Emotion task: total # of zero cell counts by cTBS condition (LPFC or S1), stimulus-discrimination accuracy (Correct or Incorrect), & PAS rating (1-4)

	Correct Trials				Incorrect Trials			
	1	2	3	4	1	2	3	4
LPFC	16	2	3	16	13	2	17	29
S1	11	0	4	13	13	3	17	28

Additionally, the Poisson test seems a bit cumbersome as it requires collapsing the scale into all possible binary scales. I was wondering whether a chi squared (goodness of fit) test would be more appropriate to check for differences in the distribution of confidence among all eight possible confidence responses (i.e. confidence of 1-4 separately for correct and incorrect responses).

We thank Reviewer 1 for this helpful suggestion! We used a Fisher's exact test instead chi-square at participant's near-threshold contrast given that the zero count for certain cells is < 5 , which is non-ideal for the latter test. We again found that the zero-cell count did not differ between LPFC & S1 cTBS conditions at participant's near-threshold contrast, orientation task Fisher's $p = 0.66$; emotion task Fisher's $p = 0.92$. We also examined the distribution of zero counts collapsed across contrasts; orientation task $\chi^2 = 0.92, p = 0.996$; emotion task $\chi^2 = 2.629, p = 0.917$.

Also, I think the name of the tests performed for these analyses should be mentioned in the manuscript as well.

They are now specified in the manuscript, as follows:

P. 28: "At participants' near-threshold contrast, zero counts did not differ between TMS conditions in either the face orientation task, Fisher's $p = .66$, or in the face emotion task, Fisher's $p = .92$ (Tables S1 & S2). Zero counts also did not differ when examined collapsed across contrasts, orientation task $\chi^2 = 0.92, p = 0.996$; emotion task $\chi^2 = 2.629, p = 0.917$."

2. The term 'metacognitive dissociations' (p6) does not seem to be a commonly used term and I was initially confused as I thought they were referring to dissociation between confidence and stimulus perception, which was not the case. Can the authors expand or rephrase this conclusion in more clear terms?

We thank the Reviewer for pointing out the ambiguity in that sentence. We have now revised it to clarify its meaning:

P. 6: "(...) dissociations between modalities of metacognition (such as low-level visual vs. emotional) may occur during the processing of complex social stimuli."

3. In Bang et al (2019), the changes in sensory noise were induced either by learning or by the mixture of trials from different levels of contrasts (which increased the between-trial sensory noise). It is not clear why sensory noise should vary between different contrasts when we use trials of the same contrast level. I wonder whether these results can be further explained by learning. If participants were able to learn during the course of the task for middle (near-threshold) contrasts, it may have lowered their sensory noise and their metacognitive efficiency. However, for contrasts much below their threshold, if no learning was possible, it could have resulted in higher sensory noise and metacognitive efficiency. I recognize that all these arguments are speculative. Nevertheless, I think that the authors' entire discussion about the variability of

metacognitive efficiency across contrasts would serve as a useful reference for future studies and should also be included in their discussion.

We thank the Reviewer for contributing to this interesting discussion and for the invitation to expand our discussion section. We have now incorporated a combination of the above-mentioned points and our prior Reply (Revision#1) into the discussion, as follows:

P. 18:

The impact of stimulus contrast on estimates of metacognitive efficiency

Metacognitive efficiency is thought to index metacognition above and beyond task performance (i.e., by subtracting stimulus discrimination performance (d') from metacognitive sensitivity measured in the same unit (meta- d') (Maniscalco & Lau, 2012). Nonetheless, in our study, metacognitive efficiency (meta- $d' - d'$) varied across face contrast (Figures 2f & 4f), exhibiting a U-shaped curve in which lowest and highest contrasts produced higher metacognitive efficiency than intermediate contrasts. Recent computational models suggest two mechanisms that may account for this pattern: (a) *Changes in sensory noise at low/intermediate contrasts*: a recent hierarchical model of confidence predicts that sensory noise produces higher estimates of metacognitive efficiency (Bang, Shekhar, & Rahnev, 2019). According to this model, metacognition (meta- d') is corrupted by both sensory and metacognitive noise, whereas stimulus discrimination performance (d') is corrupted by sensory noise only—thereby giving rise to non-constancy in metacognitive efficiency (meta- $d' - d'$) across different sensory noise levels. Perceptual learning, which reduces sensory noise, has been shown to (perhaps counter-intuitively) reduce metacognitive efficiency (meta- $d' - d'$) (Bang et al., 2019). It is possible that sensory noise varied by contrast in our study. For instance, if some degree of learning took place for near-threshold stimuli over the course of the TMS session, reduced sensory noise would account for lower metacognitive efficiency estimates at intermediate (but not lowest) contrasts¹. (b) *Error detection at high contrasts*: Theoretically, meta- $d' - d'$ should be as high as 1 (i.e., perfect metacognition)—but in practice, “hyper metacognitive sensitivity” (meta- $d' - d' > 1$) has been often observed empirically (Figure 8E in (Fleming & Daw, 2017)). This phenomenon can be explained in part by post-decisional factors, such as error detection. For instance, as stimulus strength increases (e.g., at high contrasts), errors are less frequent and more obvious, and more likely to be driven by motoric or attentional lapses (as opposed to sensory noise). The introspective observer realizes potential motoric and attentional lapses when the sensory information is clear, resulting in greater error detection. Accordingly, a Bayesian model of confidence that incorporates post-decisional factors accounts for this phenomenon: Reliable error detection gives rise to hyper metacognitive sensitivity (i.e. meta- $d' - d' > 1$) (Fleming & Daw, 2017).

¹ We thank an anonymous reviewer for this suggestion.

References

- Bang, J. W., Shekhar, M., & Rahnev, D. (2019). Sensory noise increases metacognitive efficiency. *Journal of Experimental Psychology: General*.
<https://doi.org/10.1037/xge0000511>
- Fleming, S. M., & Daw, N. D. (2017). Self-evaluation of decision-making: A general bayesian framework for metacognitive computation. *Psychological Review*, *124*(1), 91–114.
<https://doi.org/10.1037/rev0000045>
- Maniscalco, B., & Lau, H. (2012). A signal detection theoretic approach for estimating metacognitive sensitivity from confidence ratings. *Consciousness and Cognition*, *21*(1), 422–430. <https://doi.org/10.1016/j.concog.2011.09.021>